# A bone-specific adipogenesis pathway in fat-free mice defines key origins and adaptations of bone marrow adipocytes with age and disease

Xiao Zhang[1,2†], Hero Robles[1†], Kristann L Magee[1], Madelyn R Lorenz[1], Zhaohua Wang[1,3], Charles A Harris[4‡], Clarissa S Craft[1], Erica L Scheller[1,2]*

[1]Division of Bone and Mineral Diseases, Department of Medicine, Washington University, Saint Louis, United States; [2]Department of Biomedical Engineering, Washington University, Saint Louis, United States; [3]Department of Orthopaedic Surgery, Washington University, Saint Louis, United States; [4]Division of Endocrinology, Metabolism & Lipid Research, Department of Medicine, Washington University, Saint Louis, United States

**Abstract** Bone marrow adipocytes accumulate with age and in diverse disease states. However, their origins and adaptations in these conditions remain unclear, impairing our understanding of their context-specific endocrine functions and relationship with surrounding tissues. In this study, by analyzing bone and adipose tissues in the lipodystrophic 'fat-free' mouse, we define a novel, secondary adipogenesis pathway that relies on the recruitment of adiponectin-negative stromal progenitors. This pathway is unique to the bone marrow and is activated with age and in states of metabolic stress in the fat-free mouse model, resulting in the expansion of bone marrow adipocytes specialized for lipid storage with compromised lipid mobilization and cytokine expression within regions traditionally devoted to hematopoiesis. This finding further distinguishes bone marrow from peripheral adipocytes and contributes to our understanding of bone marrow adipocyte origins, adaptations, and relationships with surrounding tissues with age and disease.

*For correspondence:
scheller@wustl.edu

†These authors contributed equally to this work

Present address: ‡Regeneron Pharmaceuticals, Tarrytown, United States

Competing interest: The authors declare that no competing interests exist.

## Introduction

Bone marrow adipose tissue (BMAT) is a unique fat depot located within the skeleton. BMAT acts as an endocrine organ and energy storage depot and has the potential to contribute to the regulation of metabolism, hematopoiesis, and bone homeostasis (reviewed in *Scheller et al., 2016b*). The development and subsequent regulation of bone marrow adipocytes (BMAds) varies between skeletal sites (*Scheller et al., 2019*; *Scheller et al., 2015*; *Craft et al., 2018*; *Scheller and Rosen, 2014a*), and current work suggests that BMAds are functionally unique within the context of their niche (*Robles et al., 2019*). Specifically, the constitutive BMAT (cBMAT) begins to form in distal regions at or slightly before birth, followed by rapid expansion and maturation early in life (*Scheller et al., 2015*; *Scheller and Rosen, 2014a*). By contrast, the regulated BMAT (rBMAT) develops later and expands with age, generally in areas of active hematopoiesis (*Scheller et al., 2015*; *Scheller and Rosen, 2014a*). Recent studies in rodents and humans have also highlighted the heterogeneous metabolic properties of BMAds (*Scheller et al., 2019*; *Attané et al., 2020*), suggesting that their capacity for functional support of surrounding cells may change, particularly with age and in states of systemic disease. BMAT expansion occurs in diverse conditions including anorexia, obesity, aging, osteoporosis, hyperlipidemia, estrogen deficiency, and treatment with pharmacotherapies such as glucocorticoids and

thiazolidinediones (*Scheller et al., 2016b*; *Fazeli et al., 2013*). Many of these conditions are associated with increased fracture risk. Thus, understanding the context-specific origins and functions of BMAds has important implications for the development of clinical and pharmacologic strategies to support skeletal and metabolic health.

Genetic causes of lipodystrophy have provided clues about the molecular differences between BMAT and white adipose tissues (WAT) (reviewed in *Scheller and Rosen, 2014a*). Congenital generalized lipodystrophy (CGL) is a disorder characterized by complete loss of peripheral adipose tissues and is associated with secondary complications including hypertriglyceridemia, osteosclerosis, insulin resistance, diabetes, and hepatic steatosis (*Scheller and Rosen, 2014a*; *Fiorenza et al., 2011*; *Zou et al., 2019*; *Teboul-Coré et al., 2016*). Patients with CGL uniformly lack WAT; however, BMAT is selectively preserved in those with CGL resulting from mutations in *CAV1* (CGL3) or *PTRF* (CGL4), but not *AGPAT2* (CGL1) or *BSCL2* (CGL2) (*Scheller and Rosen, 2014a*). Similarly, all BMAT is retained in *Cav1* knockout mice, and cBMAT is present in *Ptrf* knockouts (*Scheller et al., 2015*). These results in humans and mice suggest that, unlike WAT, BMAT has unique compensatory mechanisms that promote its preservation. In this study, to define the cellular basis for this observation, we examined the formation and regulation of BMAT in the 'fat-free' (FF) *Adipoq-Cre+*, *Rosa26-lsl-DTA+* (Adipoq[Cre+/DTA+]) mouse, a novel genetic model of CGL (*Zou et al., 2019*; *Wu et al., 2018*).

In the FF mouse, any cell that expresses adiponectin (*Adipoq*-Cre+) will express diphtheria toxin A (DTA), leading to DTA-induced cell death (*Ivanova et al., 2005*; *Voehringer et al., 2008*). Adiponectin is a secreted adipokine that is expressed by all brown, white, and BMAT adipocytes in healthy mice, independent of sex (*Craft et al., 2019*). Expression of adiponectin also defines the major BMAd progenitor, termed 'Adipo-CAR' cells (adipogenic CXCL12-abundant reticular) or 'MALP' (marrow adipogenic lineage precursor) (*Zhong et al., 2020*; *Matsushita et al., 2020*; *Baccin et al., 2020*). In the FF mouse, we hypothesized that ablation of adiponectin-expressing cells would promote activation of alternate, adiponectin-negative skeletal progenitors to form adipocytes in vivo in times of systemic metabolic demand. To test this hypothesis, we performed adiponectin lineage tracing of bone marrow stromal cells and BMAds. We also analyzed age- and sex-associated changes in bone, BMAT, and peripheral adipose tissues in control and FF mice. In addition, we defined the impact of adrenergic stimulation and peripheral fat transplantation on the formation and regulation of BMAT in the FF model. Lastly, we created a triple mutant fat-free mouse with an integrated mTmG lineage reporter to explore the origins and molecular features of this unique BMAd population. This work refines our understanding of the origins and adaptations of BMAT with age and disease and defines compensatory pathways of adipocyte formation that are unique to the bone marrow and emerge in states of compromised progenitor function and altered lipid load.

## Results

### Adiponectin is expressed by BMAT adipocytes and a subset of stromal progenitor cells

As described previously (*Craft et al., 2019*), *Adipoq-Cre* (Cre+), *Rosa26-lsl-mTmG* (mTmG+) (Adipoq[Cre+/mTmG+]) lineage tracing reporter mice were used to localize adiponectin-expressing cell lineages within the skeletal niche. In this model, any cell having expressed adiponectin (*Adipoq*-Cre+) at any time during its genesis will change plasma membrane color from red to green (mT to mG, *Muzumdar et al., 2007*). Cross-sections of the proximal tibia and tail vertebrae were imaged at 3 and 16 weeks of age in both males and females after immunostaining for green fluorescent protein (GFP), red fluorescent protein (RFP), and perilipin 1 (PLIN1). Adipoq[Cre-/mTmG+] littermates were used as a negative control. In Adipoq[Cre+/mTmG+] male mice, this work confirmed that membrane-localized GFP expression was present in all PLIN1+, rBMAT adipocytes within the proximal tibia (*Figure 1—figure supplement 1A*). Prevalent GFP labeling of reticular stromal cells and bone lining cells was also noted (*Figure 1—figure supplement 1A*). GFP expression was absent in hematopoietic cells, chondrocytes, and osteocytes (*Figure 1—figure supplement 1A*). Similarly, the cells lining the endosteal bone surface were predominantly GFP/*Adipoq* negative (*Figure 1—figure supplement 1A*). In negative controls, all cells within the bone, including adipocytes and bone-lining cells, stained positive for RFP and negative for GFP (*Figure 1—figure supplement 1B*). Comparable patterns of GFP expression were observed in the proximal tibia of Adipoq[Cre+/mTmG+] female mice at both 3 and 16 weeks of age (*Figure 1—figure*

*supplement 1C* and data not shown). In the tail vertebrae, *Adipoq*-Cre traced all PLIN1+ cBMAT adipocytes independent of sex or age, as indicated by GFP (*Figure 1—figure supplement 1D*).

To determine whether adiponectin was expressed by the BMAT progenitor cell, we isolated primary bone marrow stromal cells from the femur and tibia of 16 week old male Adipoq$^{Cre+/mTmG+}$ mice for colony-forming unit (CFU) assays. After 2 weeks of expansion ex vivo, an average of 79.8 ± 9.0 % of CFUs were completely positive for adiponectin, as indicated by expression of membrane-bound GFP in 100 % of the fibroblast-appearing progenitor cells within the colony, 16.5 ± 9.1 % of CFUs were negative (RFP+ only) and 3.7 ± 0.3 % were mixed, containing both GFP+ and RFP+ fibroblasts (*Figure 1A,B*). Within these transitional colonies, cells with RFP+ membranes, indicative of their lack of adiponectin expression, routinely contained GFP+ cytoplasmic granules (*Figure 1C*). This was often near to cells that had already become fully GFP+ (*Adipoq*-Cre+), suggesting that adiponectin expression is activated at later stages of stromal progenitor maturation. Small, RFP+, myeloid-lineage cells were commonly present, particularly around the edges of the plates (*Figure 1A*). These cells did not form colonies, did not have a fibroblastic morphology, and thus were not considered in our analyses. Spontaneous adipogenesis, as indicated by the presence of PLIN1+ lipid droplets, occurred on average in 18.5 ± 1.4 % of CFUs (*Figure 1D, E*). This included 30 of the 166 total colonies examined across three independent mice. Comparable to what was observed in Adipoq$^{Cre+/mTmG+}$ mice in vivo, PLIN1+ lipid droplets were only present in GFP+ cells in vitro (*Figure 1D, E*). In negative controls, RFP+ stromal cells and PLIN1+ adipocytes were observed without the presence of GFP+ (*Figure 1F*). Together, these results suggest that all BMAds and their progenitor cells express adiponectin in healthy conditions.

## Global ablation of adiponectin-expressing cells causes sex- and age-dependent regulation of bone

Male and female FF mice were analyzed at 4 and 8 months of age relative to Adipoq$^{Cre-/DTA+}$ control littermates (Con) to isolate sex- and age-related changes in body mass, bone, and bone marrow adiposity after ablation of adiponectin-expressing cells. Male and female FF mice lacked white and brown adipose tissues and circulating adiponectin at both 4 and 8 months of age (*Figure 2A–C* and data not shown). The absence of fat was accompanied by secondary sequelae including pronounced liver enlargement and steatosis (*Figure 2B and D*) and elevated blood glucose (*Figure 2E*). Bone size, as indicated by tibia length, was reduced by 3–7% in FF male and female mice relative to controls (*Figure 2F*). Body mass was unchanged at 4 months. However, from 4 to 8 months of age, male FF mice resisted age-associated gains in body mass relative to controls (*Figure 2G*). By contrast, female FF mice were 9–13% heavier than controls at both ages examined and did not exhibit age-associated restriction (*Figure 2G*).

To assess bone morphometry, tibiae were scanned by μCT. Consistent with a previous report in younger males (*Zou et al., 2019*), trabecular bone in both male and female FF mice extended deeper into the diaphysis than controls (*Figure 3A*). In the proximal tibial metaphysis, female FF mice had increased trabecular bone volume fraction (BVF), number, thickness, and bone mineral density (BMD), with decreased trabecular spacing at both 4 and 8 months of age (*Figure 3B–F*). Increases in metaphyseal trabecular bone were less prominent in the 4 month old male FF mice (*Figure 3B*). Unlike females, trabecular number was the only factor that was increased significantly in males (*Figure 3C*) with a comparable decrease in spacing at 4 months of age (*Figure 3E*). By 8 months of age, metaphyseal trabecular BVF, BMD, number, and spacing in male FF mice were comparable to controls (*Figure 3B, C, E and F*). In addition, unlike females, male FF mice had decreased trabecular thickness relative to the control group at 8 months of age (*Figure 3D*). This reveals that ablation of adiponectin-expressing cells is sufficient to promote sustained increases in metaphyseal trabecular bone in females, but not in males.

Female FF mice at 4 months also had significantly higher cortical BVF and cortical thickness than controls (*Figure 3—figure supplement 1A-C*). Increased cortical thickness was associated with decreased medullary area and no change in total area, indicative of increased endosteal bone (*Figure 3—figure supplement 1D, E*). In male FF mice at 4 months, increased cortical BVF was also associated with decreased medullary area (*Figure 3—figure supplement 1C, D*). However, unlike in females, the total area was also decreased (*Figure 3—figure supplement 1E*), reflecting an overall decrease in bone cross-sectional size. With age, both male and female FF mice exhibited a significant

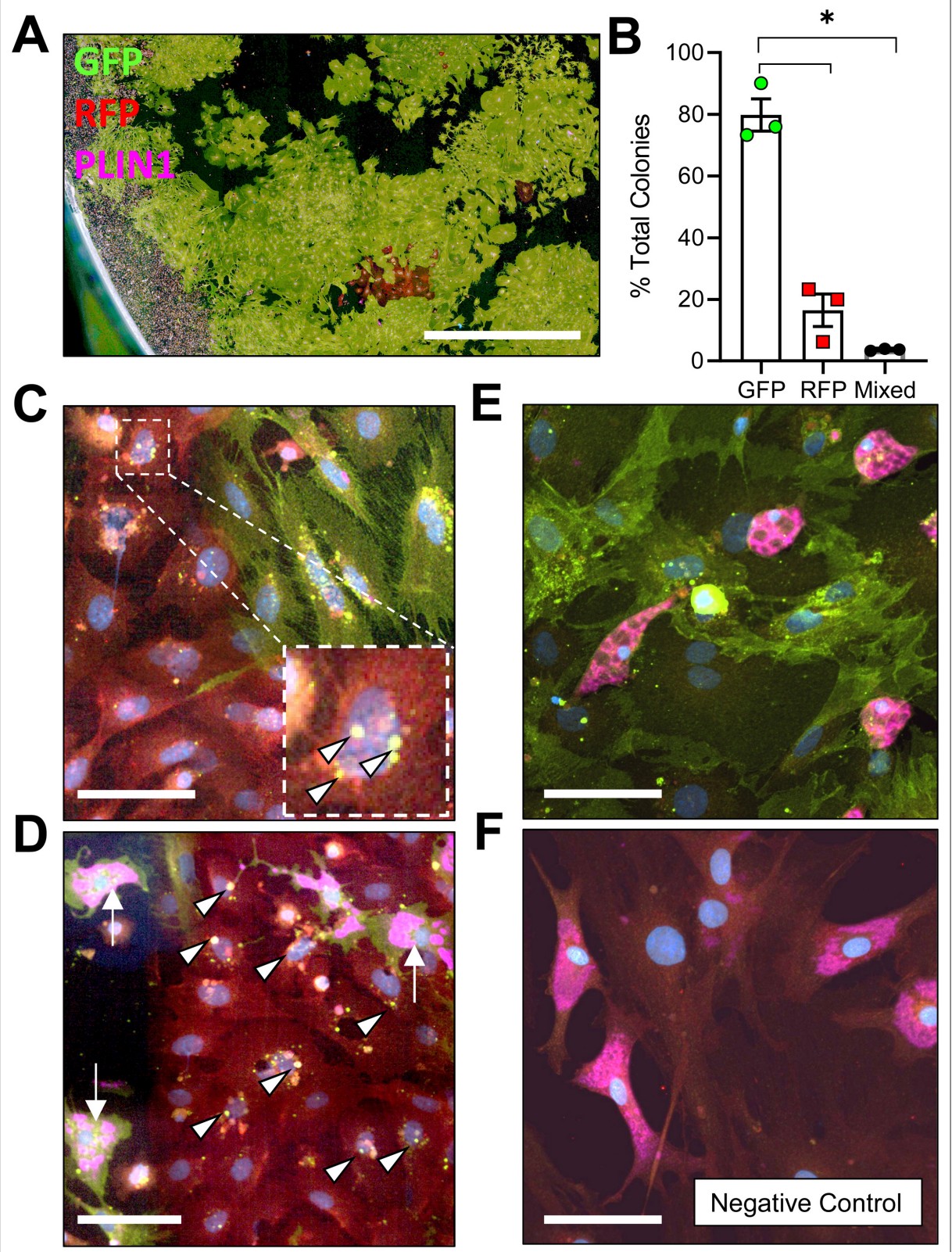

**Figure 1.** Adiponectin is expressed by Adipoq$^{Cre+/mTmG+}$ bone marrow stromal cells in vitro. (**A**) Bone marrow stromal cells from 16 week old male Adipoq$^{Cre+/mTmG+}$ mice were cultured at low density for 14 days to promote formation of colony-forming units (CFUs). Endogenous fluorescence was then amplified by immunostaining for green fluorescent protein (GFP) and red fluorescent protein (RFP). In addition, spontaneous adipogenesis was assessed based on immunostaining for perilipin 1 (PLIN1, pink). Scale = 1 mm. (**B**) Quantification of GFP and RFP expression in fibroblastic stromal cells within

*Figure 1 continued on next page*

*Figure 1 continued*

each culture dish (n = 3 independent mice, 166 total colonies counted). Data presented as mean ± SD. One-way ANOVA. *p≤0.05. (**C**) Representative mixed colony with both GFP+ and RFP+ fibroblasts demonstrating GFP+ perinuclear granules in red cells (white arrowheads), indicating upregulation of adiponectin expression. (**D**) Day 14 adipogenic colony demonstrating PLIN1+ lipid droplets (pink, white arrows) in GFP+ adipocytes. Nearby RFP+ fibroblasts show early signs of conversion (GFP+ perinuclear granules, white arrowheads). (**E**) Day 14 adipogenic colony with uniformly GFP+ stromal cells and PLIN1+ adipocytes. (**F**) Adipoq^{Cre-/mTmG+} negative control has RFP+ bone marrow stromal cells and RFP+, PLIN1+ adipocytes. (**C–F**) Scale = 20 µm.

The online version of this article includes the following source data and figure supplement(s) for figure 1:

**Source data 1.** Percent total colony count.

**Figure supplement 1.** Adiponectin is expressed by BMAT adipocytes in the Adipoq^{Cre+/mTmG+} mouse in vivo.

–19.0 % and –19.1 % decrease in tibial cortical bone thickness, respectively (***Figure 3—figure supplement 1B***). This was in direct contrast with control mice, where tibial cortical thickness remained constant (male) or was increased by +16 % (female) with age (***Figure 3—figure supplement 1B***). Changes in the bone mineral content (BMC) mirrored this result, with age-associated increases in controls, but not in FF mice (***Figure 3—figure supplement 1G***). There were no significant differences in predicted torsional bone strength by polar moment of inertia at any of the ages examined (***Figure 3—figure supplement 1H***). Overall, this demonstrates that ablation of adiponectin-expressing cells promotes early gains in the amount and thickness of cortical bone; however, these increases are not sustained and tend to be normalized or decreased relative to controls with age.

## Global ablation of adiponectin-expressing cells drives ectopic expansion of BMAT

Tibiae from the 4 and 8 month old FF and control mice were decalcified and stained with osmium tetroxide for visualization and quantification of BMAT. Unlike peripheral adipose tissues, the 3D-reconstructed images of the osmium-stained tibiae indicated that BMAT was still present (***Figure 4A and B***). When quantified and expressed relative to total bone marrow volume, the percentage total tibial BMAT was comparable to controls in 4 month old FF male and female mice and in 8 month old males (***Figure 4C***). In control mice, BMAT was localized in the well-established pattern of concentration within proximal and distal ends of the tibia (***Figure 4A and B***; ***Scheller et al., 2015***). By contrast, the BMAT in the FF mice was found predominantly in the proximal tibia and mid-diaphyseal region with few adipocytes in the distal tibia (***Figure 4A and B***). Consistent with the 3D reconstructions, regional sub-analyses revealed that retained BMAT adipocytes were primarily localized proximal to the tibia/fibula junction (***Figure 4D***). Within the proximal tibia, BMAT increased by 2.2-fold in control males and 5.6-fold in control females from 4 to 8 months of age (***Figure 4D***). In FF mice, though the absolute volume of BMAT was similar or less than controls (***Figure 4D***), proximal tibial BMAT increased by 6.9-fold and 23.2-fold with age in males and females, respectively (***Figure 4D***). This included expansion within the mid-diaphysis, a region in mice that generally has very low amounts of BMAT (***Figure 4B***). In the distal tibia, control males and females had a large volume of BMAT at 4 months that also increased by 2.7- and 2.3-fold with age (***Figure 4E***). By contrast, FF mice had very little BMAT in the distal tibia and, though minor increases with age were noted, these changes were not significant (***Figure 4E***). Distal tibia BMAT often behaves similarly to constitutive BMAT in regions such as the tail vertebrae (***Scheller et al., 2019***; ***Scheller et al., 2015***). Consistent with this, BMAT adipocytes were absent in the 8-month-old FF tail vertebrae, a region of dense cBMAT-like adipocytes in control mice (***Figure 4F***). These findings demonstrate that BMAT persists in FF mice despite global ablation of adiponectin-expressing cells and, further, that these ectopic BMAT adipocytes expand with age primarily in regions traditionally comprised of red bone marrow.

By histology, FF BMAds in the tibia and femur were morphologically comparable to control BMAT adipocytes (***Figure 5A and B***). FF BMAds contained a large, central PLIN1+ lipid droplet (***Figure 5B and C***) and were negative for macrophage-marker CD68 (***Figure 5B***). Histologic sections also confirmed the DTA-mediated depletion of the peripheral peri-skeletal adipose tissues in FF mice (***Figure 5A***). For example, control mice had infrapatellar PLIN1+ adipocytes in the knee joint region (***Figure 5A and B***). By contrast, in FF mice, the infrapatellar adipocytes were replaced with a population of foam-cell like auto-fluorescent, PLIN1–, CD68+ macrophages (***Figure 5A and B***). The same

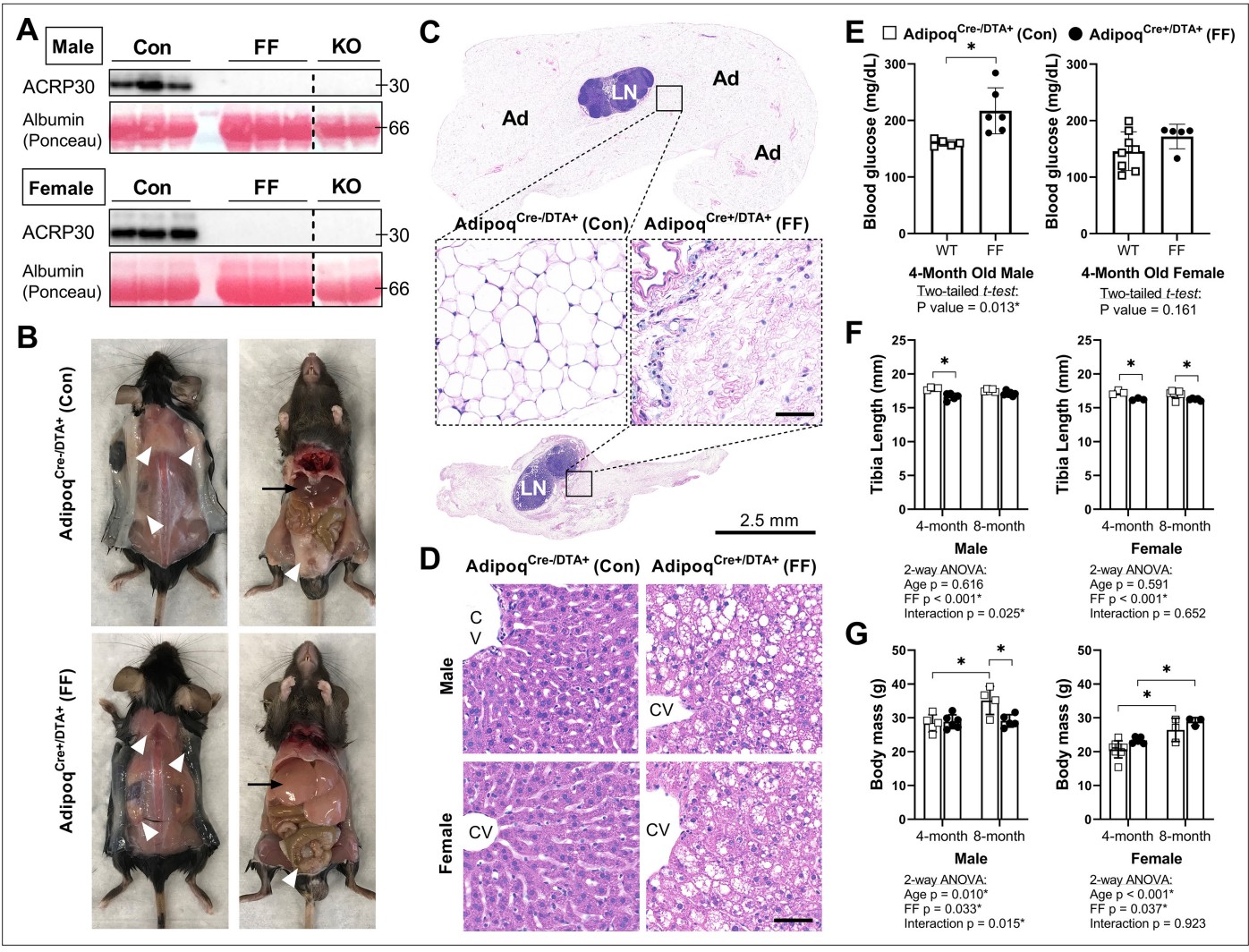

**Figure 2.** Adipoq[Cre+/DTA+] fat free (FF) mice lack white and brown adipose tissues and circulating adiponectin. (**A**) Serum adiponectin (ACRP30) of Adipoq[Cre-/DTA+] control (Con), Adipoq[Cre+/DTA+] fat free (FF), and Adipoq knockout (KO) mice by western blot. Blood albumin levels by Ponceau S staining (loading control). (**B**) Representative pictures showing the absence of white and brown adipose tissues (white arrowheads) and the enlarged liver (black arrows) in 16 week old male FF mice relative to control. Identical gross phenotypes were observed in females (data not shown). (**C**) Representative hematoxylin and eosin (H&E) stained sections of inguinal white adipose tissue. Areas of adipocytes have been replaced by loose fibrous tissue in FF mice. Ad = adipocytes. LN = lymph node. Inset scale = 50 μm. (**D**) Representative (H&E) stained sections of liver. CV = central vein. Scale = 50 μm. (**E**) Random fed blood glucose, measured using a glucometer. (**F**) End point tibia lengths, measured using a caliper. (**G**) Body mass. Sample size for control and FF mice, respectively: 4 months Male n = 5, 6, Female n = 8, 5; and 8 months Male n = 4, 5; Female n = 3, 3. (E) Two-tailed t-test, (**F, G**) two-way ANOVA with Tukey's multiple comparisons test. ANOVA results as indicated. *p≤0.05. Data presented as mean ± SD. WT and FF mice were housed at 30 °C on a 12 hr/12 hr light/dark cycle.

The online version of this article includes the following source data for figure 2:

**Source data 1.** Blood glucose, tibia length, and body mass.

**Source data 2.** Western blot uncropped images.

result was observed in the extra-skeletal adipose tissues surrounding the tail vertebrae and the bones in the feet (*Figure 5—figure supplement 1*). This confirms that adipocyte cell death occurs uniformly in the peripheral fat tissues, with selective adipocyte preservation within the bone marrow of FF mice. FF BMAT adipocytes were on average 13.7% and 42.9% larger than controls in male and female mice, respectively, reflecting increases in lipid storage (*Figure 5D*).

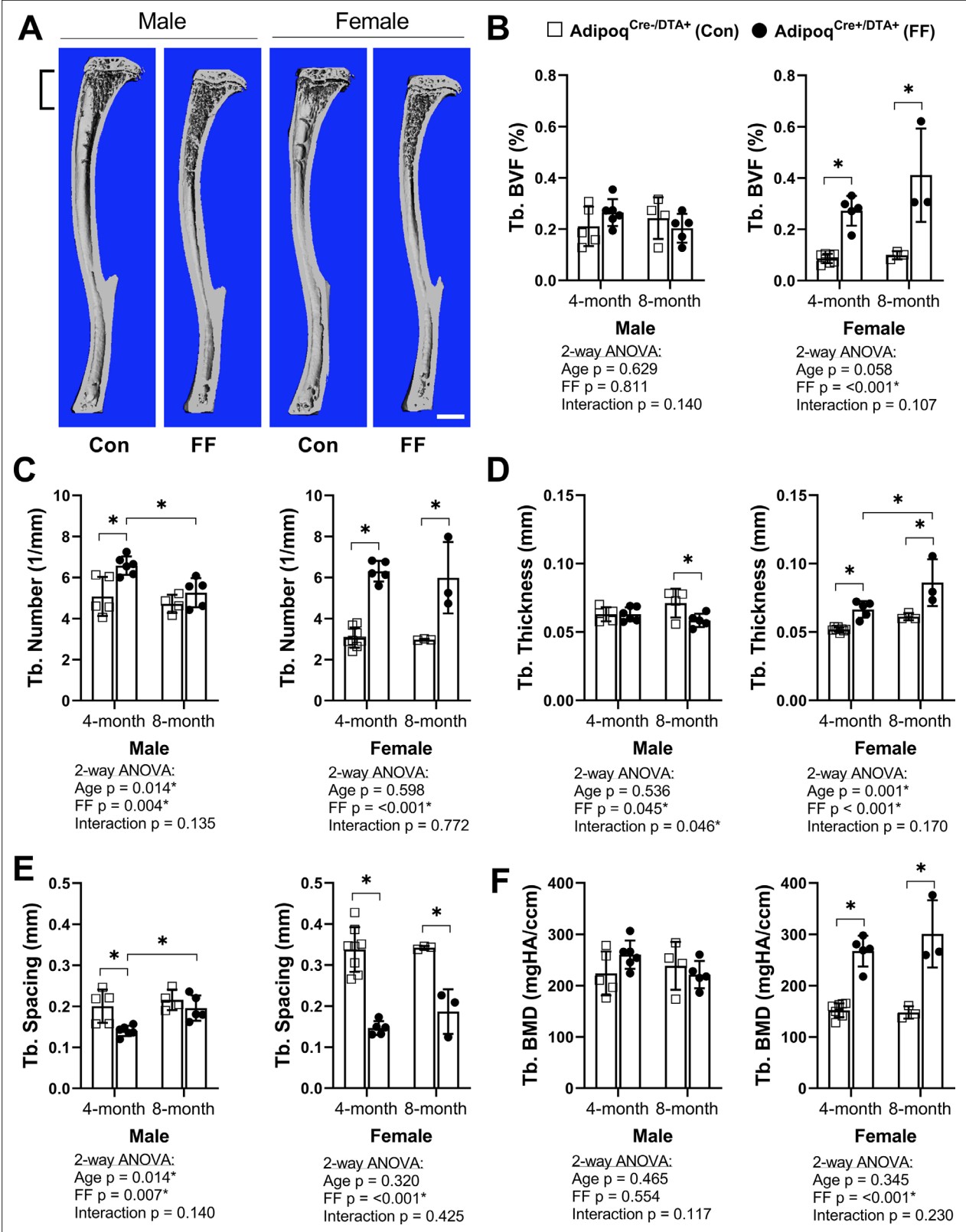

**Figure 3.** Trabecular bone is increased in Adipoq[Cre+/DTA+] fat free (FF) mice. (**A**) Representative 3D µCT-based reconstructions of tibiae from 4 month old Adipoq[Cre+/DTA+] fat free (FF) and Adipoq[Cre-/DTA+] controls (Con). Scale = 1 mm. (**B–F**) Quantification of trabecular parameters in the proximal tibial metaphysis. Region of interest as indicated in (**A**). (**B**) Trabecular bone volume fraction (Tb. BVF). (**C**) Trabecular number. (**D**) Trabecular thickness. (**E**) Trabecular spacing. (**F**) Trabecular bone mineral density (Tb. BMD). Sample size for control and FF mice, respectively: 4 months Male n = 5, 6, Female

*Figure 3 continued on next page*

*Figure 3 continued*

n = 8, 5; and 8 months Male n = 4, 5; Female n = 3, 3. Statistical significance was assessed by two-way ANOVA with Tukey's multiple comparisons test. ANOVA results as indicated. *$P \leq 0.05$. Data presented as mean ± SD. WT and FF mice were housed at 30 °C on a 12 hr/12 hr light/dark cycle.

The online version of this article includes the following source data and figure supplement(s) for figure 3:

**Source data 1.** Trabecular bone parameters.

**Figure supplement 1.** Cortical parameters are increased in young Adipoq^Cre+/DTA+ fat free (FF) mice, but tends to decline with age.

**Figure supplement 1—source data 1.** Cortical bone parameters.

## Ectopic BMAT in FF mice is not regulated by cold stress or $\beta$3-adrenergic stimulation

Regulation of BMAT adipocytes by adrenergic stimulation has important implications for the functional integration of BMAds with local and peripheral energy stores. FF mice lack WAT and BAT and have impaired thermoregulatory capabilities. Thus, control and FF mice are bred and housed at

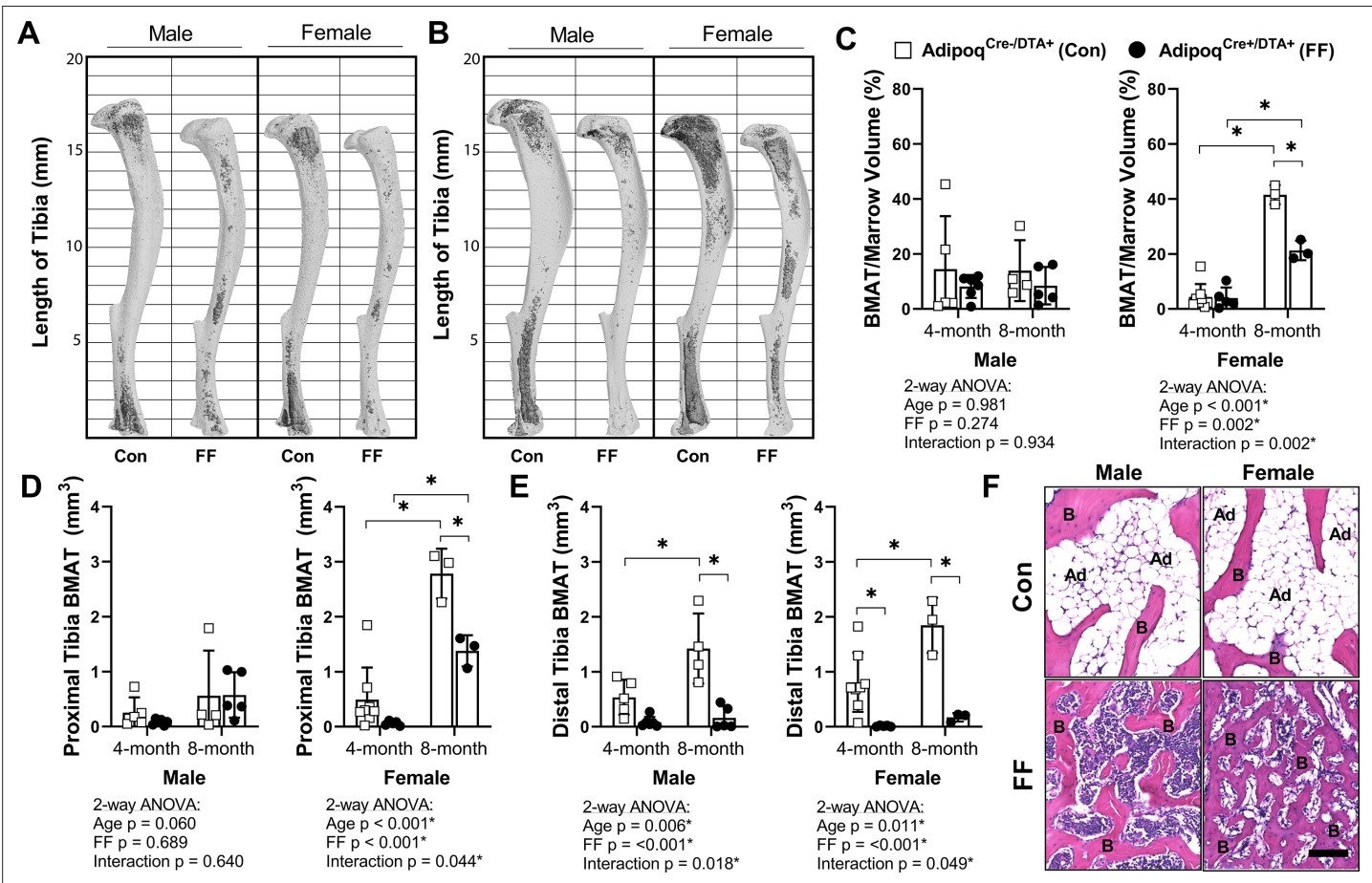

**Figure 4.** BMAT is present in Adipoq^Cre+/DTA+ fat free (FF) mice and expands with age. (**A, B**) Representative µCT images of osmium-stained tibiae for both male and female Adipoq^Cre+/DTA+ fat free (FF) and Adipoq^Cre-/DTA+ control (Con) mice at (**A**) 4 months and (**B**) 8 months of age. BMAT is in dark grey and bone is in light gray. (**C**) Quantification of total tibial BMAT volume as a percentage of total bone marrow volume. (**D**) Regional analysis of BMAT within the proximal end of the same tibiae as in (**C**), expressed as the total volume of osmium-stained lipid from the proximal end of the tibia to the tibia/fibula junction. (**E**) Regional analysis of BMAT within the distal end of the same tibiae as in (**C**), expressed as the total volume of osmium-stained lipid from tibia/fibula junction to the distal end of the bone. (**F**) Representative hematoxylin and eosin (H&E) stained sections of BMAT within tail vertebrae. Ad = BMAT adipocytes. B = bone. Scale = 50 µm. Sample size for control and FF mice, respectively: 4 months Male n = 5, 6, Female n = 8, 5; and 8 months Male n = 4, 5; Female n = 3, 3. Statistical significance was assessed by two-way ANOVA with Tukey's multiple comparisons test. ANOVA results as indicated. *p≤0.05. Data presented as mean ± SD. WT and FF mice were housed at 30 °C on a 12 hr/12 hr light/dark cycle.

The online version of this article includes the following source data for figure 4:

**Source data 1.** BMAT volume.

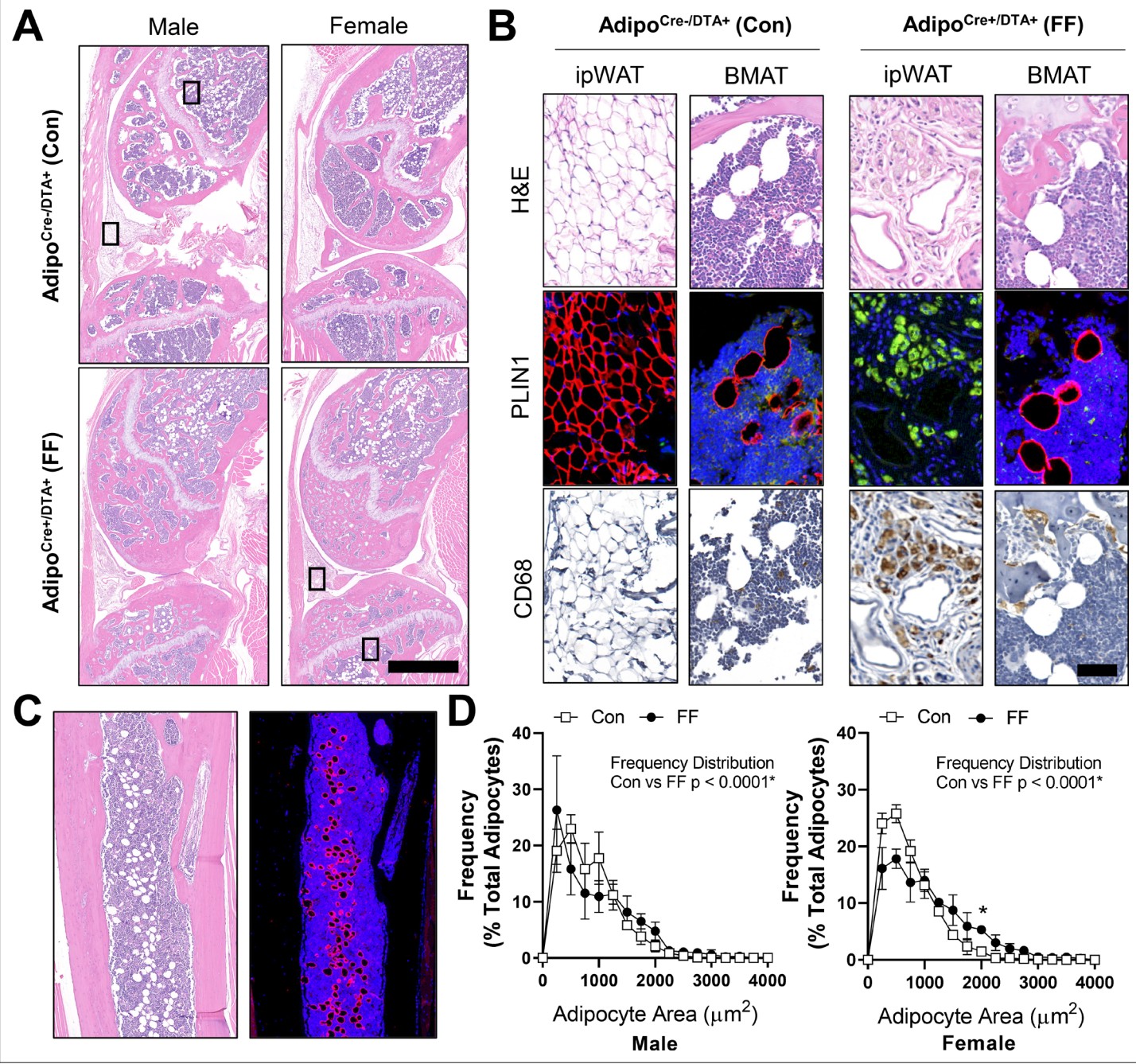

**Figure 5.** Fat free (FF) bone marrow adipocytes express perilipin, but not CD68, and have increased lipid storage relative to controls. Representative images from both male and female Adipoq^Cre+/DTA+ fat free (FF) and Adipoq^Cre-/DTA+ control (Con) mice at 8 months of age. (**A**) Representative longitudinal hematoxylin and eosin (H&E) stained sections of the femur and tibia, including the knee and surrounding soft tissue. Scale = 1 mm. (**B**) Representative serial sections stained with H&E, perilipin 1 (PLIN1, red; DAPI, blue), and CD68 (amplified with DAB, CD68+ cells are brown). Sections from the insets depicted in (**A**). ipWAT = infrapatellar white adipose tissue, located within the knee joint. BMAT = bone marrow adipose tissue. Scale = 50 μm. (**C**) Representative serial sections of the ectopic adipocytes within the tibial diaphysis in the Adipoq^Cre+/DTA+ (DTA) mice. Stained with H&E (left) and perilipin 1 (PLIN1, red; DAPI, blue). (**D**) Bone marrow adipocyte size distribution in the proximal tibia of the control and FF mice at 8 months of age. Scale = 500 μm. Sample size for control and FF mice, respectively: 8 months Male n = 4, 5; Female n = 3, 3. Statistical significance was assessed by two-way ANOVA with Tukey's multiple comparisons test. ANOVA results as indicated. Data presented as mean ± SD. *p≤0.05. WT and FF mice were housed at 30 °C on a 12 hr/12 hr light/dark cycle.

The online version of this article includes the following source data and figure supplement(s) for figure 5:

**Source data 1.** BMAd size distribution.

**Figure supplement 1.** Extra-skeletal adipocytes are depleted in the foot and tail of FF mice.

thermoneutrality (30 °C). To assess the response of FF bone and BMAds to thermal stress, male control and FF mice were housed at thermoneutrality (30 °C) or room temperature (22 °C) for 3–4 months, beginning at 4 weeks of age. Under mild cold stress (22 °C), trabecular BVF was decreased by 33–57% relative to housing at thermoneutrality (30 °C) in the tibia and femur of control and FF mice (*Figure 6A*). By comparison, cortical thickness was 9–13% lower in control mice at 22 °C but remained unchanged in FF mice, regardless of temperature or analysis site (*Figure 6B*). By osmium µCT, BMAT in the proximal tibia was 82 % lower in control mice housed at 22 °C than in mice housed at 30 °C (*Figure 6C and D*). However, similar to cortical bone, proximal tibial BMAT remained unchanged with mild cold stress in FF mice (*Figure 6C and D*). Retention of BMAT in the FF mice was also prevalent in the femur and presented with the same atypical pattern of accumulation in the mid-diaphysis (*Figure 6E*). Regulation of BMAT within the femur mirrored that observed in the proximal tibia, though it did not reach statistical significance (*Figure 6E and F*). Unexpectedly, BMAT in both control and FF mice in the distal tibia increased by 1.5- and 5-fold, respectively, in mice housed at 22 °C (*Figure 6C and D*). Overall, this result indicates that in control mice, trabecular bone, cortical bone, and BMAT are decreased in response to mild cold stress. By contrast, only trabecular bone is regulated by thermal stress at 22 °C in the FF mice with no observed changes in cortical bone or BMAT.

Next, to isolate responses of control and FF BMAT to direct β-adrenergic stimulation, we treated 7.5 month old male mice with CL316,243, a β3-adrenergic receptor (β3-AR) agonist. Eight daily subcutaneous injections of CL316,243 were administered over the course of 10 days (weekdays only, Monday to Friday in week 1 followed by Monday to Wednesday in week 2) prior to sacrifice on Day 11. To monitor the efficacy of the CL316,243 over time, circulating glycerol concentrations were measured on day 1 and day 7 both immediately prior to and 30 min after the CL316,243 injection. Increases in circulating glycerol occur secondary to activation of adipocyte lipolysis and triglyceride hydrolysis by β3-AR (*Scheller et al., 2019*; *Mottillo et al., 2014*). In control mice, CL316,243 evoked a 2.2-fold and 2.3-fold increase in circulating glycerol on days 1 and 7, respectively (*Figure 7A*). This response to CL316,243 was absent in FF mice (*Figure 7A*). After 10 days, β3-AR stimulation decreased BMAT adipocyte cell area in the proximal tibia by 26 % in control mice (*Figure 7B and C*). This reflects an estimated 37 % decrease in adipocyte cell volume (*Estimated Volume* = $\frac{4}{3}\pi r^3$, 3D, µm³). BMAd size was unchanged by β3-AR stimulation in FF mice (*Figure 7B and C*). Expression of β3 adrenergic receptor (*Adrb3*) and monoglyceride lipase (*Mgll*) were significantly decreased in purified BMAds from FF mice relative to controls (*Figure 7D*). By contrast, gene expression of β2 adrenergic receptor (*Adrb2*), adipose triglyceride lipase (*Pnpla2*), and hormone-sensitive lipase (*Lipe*) were comparable in FF BMAds (*Figure 7D*). Together, these results demonstrate that the ectopic BMAT in FF mice is resistant to cold and β3-AR agonist-induced lipolytic stimulation and that the retained BMAds have underlying deficits in β3-adrenergic receptor and monoglyceride lipase expression that may explain this resistance to lipolysis.

## Subcutaneous fat transplant prevents ectopic BMAT expansion in FF mice

We hypothesized that BMAT expansion in FF mice occurs secondary to peripheral fat depletion and hypertriglyceridemia. To test this hypothesis, male and female FF and control mice underwent sham surgery or were transplanted subcutaneously with wild-type adipose tissue at 3–5 weeks of age. After surgery, mice were monitored for 12 weeks prior to euthanasia. There were no differences in the body mass of the male mice over time regardless of genotype or transplant (*Figure 8A*). In females, consistent with previous 4 and 8 month old cohorts (*Figure 2G*), the body mass of the non-transplanted FF mice was 14–16% higher, on average, than controls (*Figure 8A*). Subcutaneous fat transplant reduced the body mass of the FF female mice to control levels (*Figure 8A*). Fat transplant was also sufficient to normalize the hyperglycemia present in both the male and female FF animals (*Figure 8B*). At the endpoint, the total mass of the transplanted adipose tissue was significantly higher in the FF mice than in the controls (*Figure 8C*). As has been reported previously (*Zou et al., 2019*), engrafted adipose tissue fragments resembled subcutaneous white adipose tissue at the time of sacrifice (*Figure 8—figure supplement 1*). There was also a significant rescue of liver enlargement and peripheral hypertriglyceridemia by fat transplant in both male and female FF mice (*Figure 8D and E*). Fat transplant did not substantially modify the cortical and trabecular bone phenotypes in the tibia (*Figure 8—figure supplement 2*). However, fat transplant was sufficient to prevent the ectopic BMAT expansion in FF

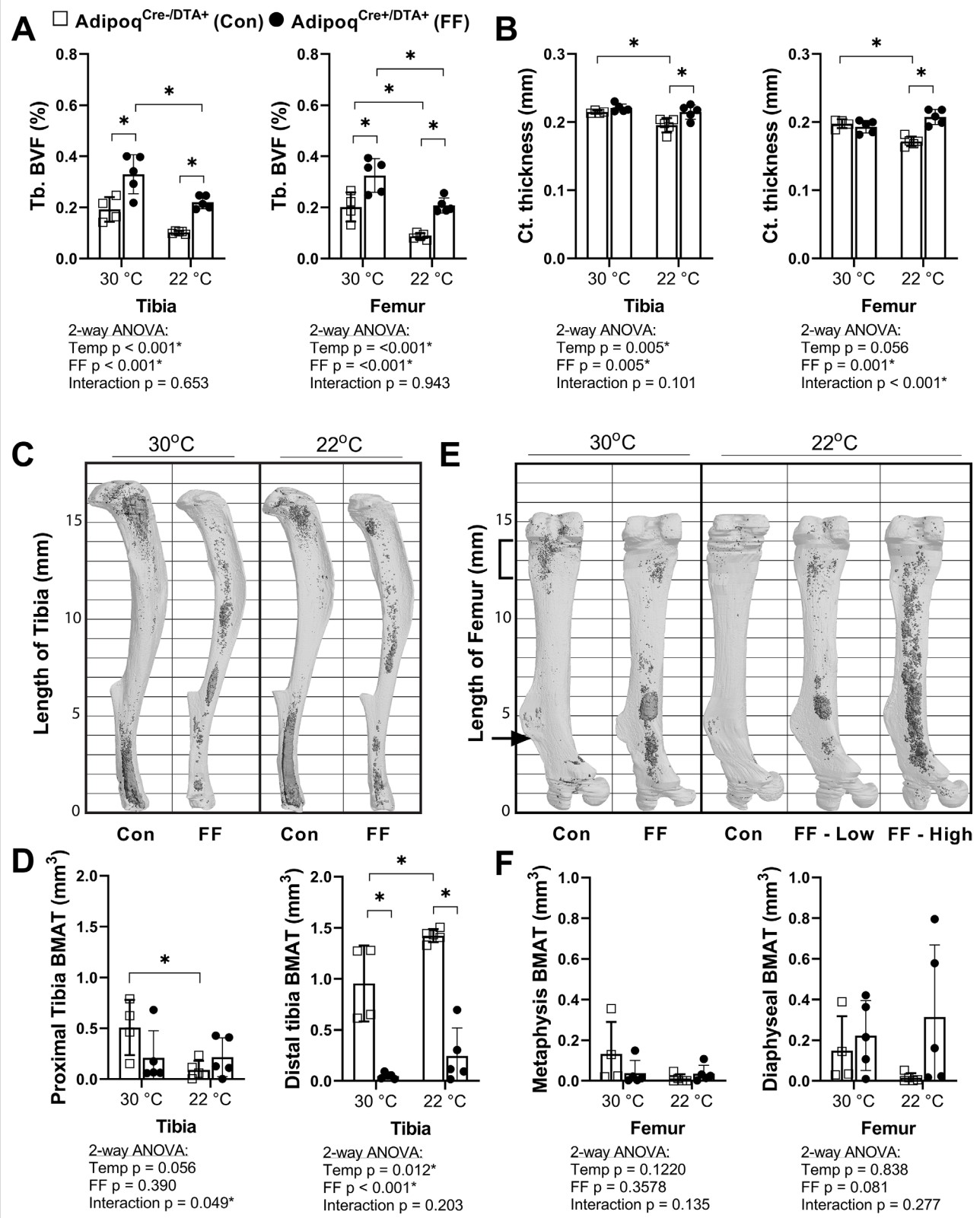

**Figure 6.** BMAT in Adipoq^Cre+/DTA+ fat free (FF) mice is not responsive to cold temperature challenge (22 °C vs 30 °C). Male Adipoq^Cre+/DTA+ FF mice and controls were maintained in thermoneutral housing (30 °C) or moved to room temperature (22 °C) at 3–5 weeks of age. Bones were analyzed after 3–4 months, at 15–17 weeks of age. (**A**) Trabecular bone volume fraction (Tb. BVF) of tibia and femur. (**B**) Cortical thickness of tibia and femur analyzed by μCT. (**C**) Representative μCT images of osmium-stained tibiae at endpoint, respectively. Bone marrow fat is in dark grey and bone is in light gray.

*Figure 6 continued on next page*

*Figure 6 continued*

(**D**) Quantification of osmium-stained BMAT in the region proximal to the tibia/fibula junction (proximal tibia) or distal to this point (distal tibia). (**E**) Representative µCT images of osmium-stained femur at end point, respectively. (**F**) Quantification of osmium-stained BMAT in the 2 mm region below the growth plate (femur metaphysis, bracket) or from this point to the end of the femur flange (indicated by the arrow, diaphyseal BMAT). Sample size of control and FF, respectively: 30 °C n = 4, 5; 22 °C n = 5, 5. Statistical significance was assessed by two-way ANOVA with Tukey's multiple comparisons test. ANOVA results as indicated. *p≤0.05. Data presented as mean ± SD.

The online version of this article includes the following source data for figure 6:

**Source data 1.** Cold challenge source data.

mice (*Figure 8F–H*). An independent increase in tibial BMAT volume was also observed in male WT fat transplanted mice (*Figure 8F and H*). The reason for this is unclear as no differences were noted in BAT, iWAT, or gWAT mass after fat transplant in male or female control mice (*Figure 8—figure supplement 3A-C*). Overall, these results reinforce the critical role of the peripheral adipose tissue as a lipid storage depot that reduces the systemic burden of hypertriglyceridemia on peripheral tissues such as liver and bone marrow.

## Origins and adaptations of ectopic BMAT adipocytes

To further determine the origin of ectopic BMAds in FF mice while testing the efficacy of the DTA, we generated triple-mutant Adipoq^Cre+/DTA+/mTmG+ lineage tracing reporter mice (FF^mTmG). Adipoq^Cre-/DTA+/^mTmG+ mice were used as negative controls (Con^mTmG). In the FF^mTmG model, *Adipoq*-expressing cells will express GFP and should undergo DTA-mediated cell death. By contrast, cells that do not express *Adipoq* will express RFP. To begin, cross-sections of femur and tibia from male and female FF^mTmG at 4 months of age were imaged after immunostaining for GFP, RFP, and PLIN1 to assess adiponectin expression in the ectopic BMAds. Unexpectedly, both adiponectin-negative (RFP+, 41–56%) and adiponectin-expressing (GFP+, 6–21%) BMAT adipocytes were present in FF^mTmG mice (*Figure 9A and B*). In addition, approximately 38 % of cells had indeterminate membranes, with evidence of both RFP and GFP expression in vivo (*Figure 9A and B*). Next, we plated primary bone marrow stromal cells (BMSCs) from Con^mTmG and FF^mTmG mice for quantitative CFU assays (*Figure 9C*). Consistent with the broad expression of *Adipoq* in the stromal compartment (*Figure 2*, *Supplementary file 1*), the number of CFUs was reduced by 91 % in FF^mTmG mice (*Figure 9D*). In addition, in line with our in vivo results, spontaneous adipogenesis in the residual FF^mTmG BMSCs was observed in both adiponectin-negative (RFP+) and adiponectin-expressing (GFP+) stromal progenitor cells in vitro (*Figure 9C*). This result suggests that the majority of FF BMAds originate from adiponectin-negative stromal cells, with a minor portion of BMAds being adiponectin positive. This minor population of cells was not sufficient to restore circulating adiponectin, as adiponectin levels in FF mice were not increased when compared to adiponectin knockout animals (*Figure 2A*).

Next, we analyzed the gene expression of purified BMAds from control and FF mice. As expected, expression of *Adipoq* was decreased in BMAds from FF mice (*Figure 9E*). In addition, expression of cytokines including stromal cell-derived factor 1, also known as C-X-C motif chemokine 12 (*Cxcl12*), adipsin (*Cfd*), and resistin (*Retn*), were significantly decreased (*Figure 9E*). Expression of adipogenic transcription factor peroxisome proliferator-activated receptor gamma (*Pparg*) was also decreased. By contrast, expression of CCAAT/enhancer-binding protein alpha (*Cebpa*), fatty acid transporter *Cd36*, and alkaline phosphatase (*Alpl*) were comparable in control and FF BMAds. Lastly, we assessed the expression of diphthamide biosynthesis enzymes 1–7 (*Dph1-7*), as deficiency in even a single *Dph* enzyme can confer resistance to DTA-mediated cell death (*Liu et al., 2004*; *Uthman et al., 2013*). We did not observe significant regulation of any *Dph* genes in FF BMAds relative to control BMAds (*Figure 9F*). Overall, our findings define the FF BMAd as an ectopically positioned, PLIN1+, CD68− adipocyte that is specialized for lipid storage with decreased capacity for lipid mobilization and expression of cytokines including adiponectin, resistin, adipsin, and *Cxcl12*.

## Discussion

It has previously been assumed that all adipocytes, including BMAds, express the adipokine adiponectin (*Craft et al., 2019*; *Scheller et al., 2016a*; *Cawthorn et al., 2014*) and, conversely, that adiponectin is not expressed by cells that are not adipocytes. However, recent lineage tracing and single-cell

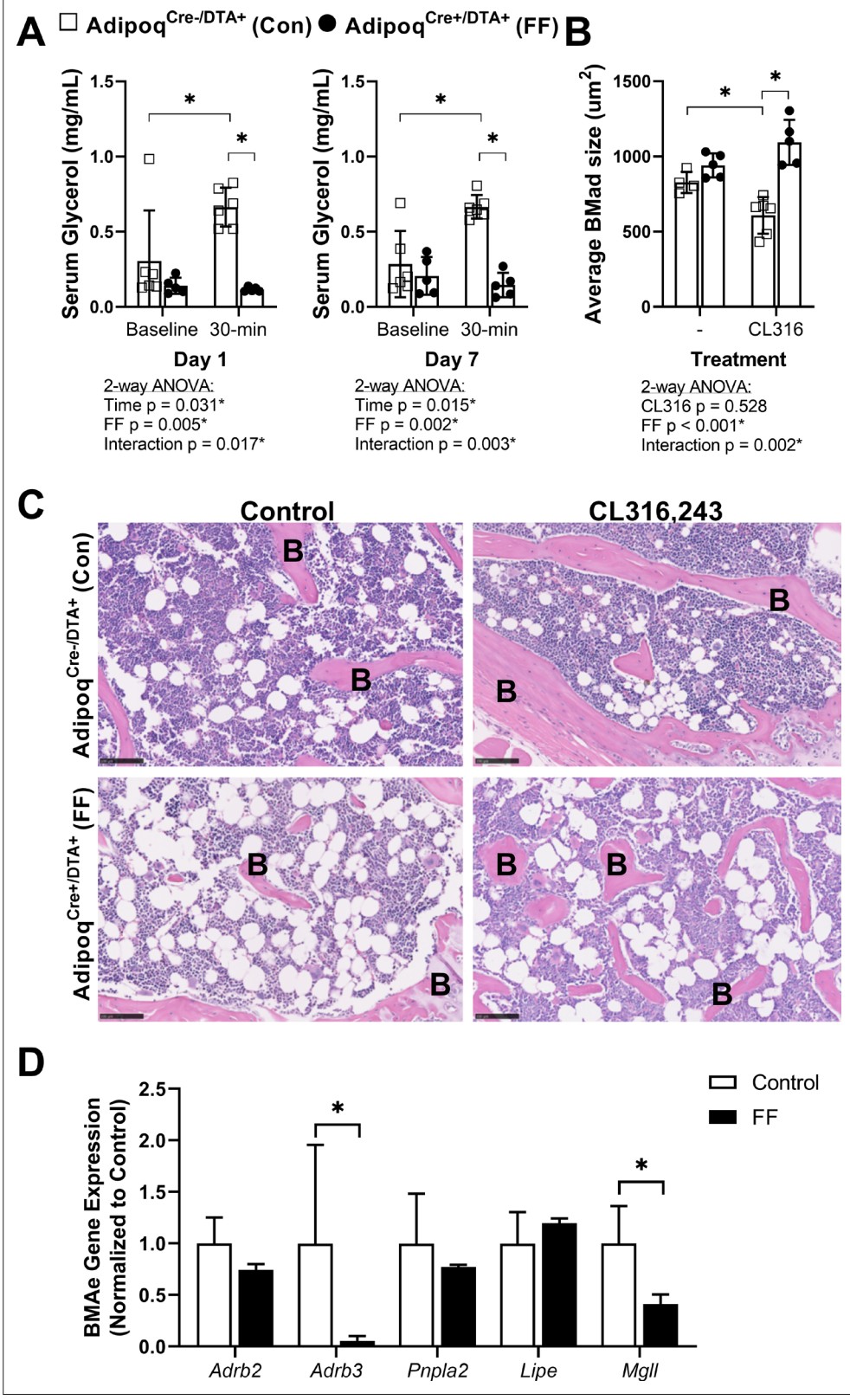

**Figure 7.** BMAT in Adipoq<sup>Cre+/DTA+</sup> fat free (FF) mice is not regulated by β3-adrenergic stimulation. Male Adipoq<sup>Cre+/DTA+</sup> FF and Adipoq<sup>Cre-/DTA+</sup> controls (Con) were treated with CL316,243, a β3-adrenergic receptor (β3-AR) agonist, using a new chronic treatment regimen. Eight daily subcutaneous injections of 0.03 mg/kg CL316,243 were administered to 7.5 month old control and FF mice over the course of 10 days (weekdays only, M to F, M to W)

*Figure 7 continued on next page*

*Figure 7 continued*

prior to sacrifice on Day 11. (**A**) Serum glycerol at days 1 and 7 of the treatment regimen. (**B**) Average adipocyte cell size in the proximal tibia as assessed in ImageJ using H&E stained slides. (**C**) Representative H&E stained sections. Sample size of control and FF, respectively: 8 month old non-treatment control n = 4, 5 (same mice as in *Figures 3–5*), 8 month old CL316,243 treated n = 6, 5. Statistical significance was assessed by two-way ANOVA with Tukey's multiple comparisons test. ANOVA results as indicated. (**D**) Gene expression of the indicated targets normalized to the geometric mean of housekeeping genes Ppia and Tbp in floated cell preparations enriched for bone marrow adipocytes (BMAe), each gene expressed relative to its respective control. Control n = 2–4, representative of pooled samples from 20 to 37 mice; FF n = 2, representative of pooled samples from 20 mice. Unpaired t-test with Holm–Sidak correction for multiple comparisons. Data presented as mean ± SD. *p≤0.05. All mice were housed at 30 °C on a 12 hr/12 hr light/dark cycle.

The online version of this article includes the following source data for figure 7:

**Source data 1.** Serum glycerol, average BMAd size, and BMAe gene expression.

RNAseq studies, including the data presented here, challenge this paradigm and demonstrate that adiponectin is expressed by a subset of bone marrow stromal progenitor cells. These adiponectin-expressing progenitors overlap with CAR cells (*Matsushita et al., 2020*; *Baccin et al., 2020*; *Mukohira et al., 2019*) and have been more recently termed MALPs (*Zhong et al., 2020*). They are largely positive for PDGFRβ and VCAM-1 and have a unique gene expression pattern that mimics known features of pre-adipocytes (*Matsushita et al., 2020*; *Baccin et al., 2020*; *Mukohira et al., 2019*). Ex vivo culture of primary bone marrow stromal cells from Adipoq[Cre+/mTmG+] reporter mice in this study further demonstrated their ability to differentiate into mature adipocytes (*Figure 1*), supporting these previous findings. Adiponectin-expressing stromal progenitors appear after birth (P1+), matching the known development of BMAT which also occurs primarily postnatally (*Scheller et al., 2015*; *Scheller and Rosen, 2014a*). Consistent with this and likely also due to the high expression and secretion of adiponectin by healthy BMAds (*Cawthorn et al., 2014*), classical depots of rBMAT and cBMAT failed to form in the Adipoq[Cre+/DTA+] FF mouse (*Scheller et al., 2015*). However, instead, an ectopic population of FF BMAds developed with age in regions of the skeleton such as the diaphysis that are generally devoid of BMAT.

## Ectopic FF BMAds originate from adiponectin[-/low] progenitors and express low levels of Cxcl12

In triple-mutant Adipoq[Cre/DTA+/mTmG+] lineage tracing reporter mice, 41–56% of BMAds were RFP+, demonstrating that at least half of the FF BMAds originated from an adiponectin-negative progenitor population. A second subset of cells appeared to have mixed or indeterminate membranes (~38%) and a final population was GFP positive (6–21%). The reason for this is unclear, as expression of *Dph1-7* genes did not indicate a clear mechanism for DTA resistance. It may be that, with time, the adiponectin-negative BMAds gradually evolved to express a low level of adiponectin. If these cells are continually forming and subsequently being cleared by DTA, this may explain why we observed some GFP+ cells in our analyses. Regardless, as a population, this unique ectopic BMAd population did not rescue circulating adiponectin in the FF mice and had decreased expression of *Adipoq*.

The location of the adiponectin[-/low] FF BMAds aligns with known sites of arteriolar entry and distribution within the femur and tibia (*Grüneboom et al., 2019*; *Asghar et al., 2020*). These cells also expressed *Cxcl12*, though this was decreased relative to control BMAds. Arterioles have recently been defined as a site of Osteo-CAR cells, a subpopulation of *Cxcl12*-expressing cells that are enriched for osteogenic progenitors, while Adipo-CAR or MALP cells are primarily localized to the venous sinusoids (*Zhong et al., 2020*; *Baccin et al., 2020*). Additional adiponectin-negative, *Cxcl12*-negative mesenchymal progenitor populations also exist within bone (*Chen et al., 2020*; *Kusumbe et al., 2016*; *Ramasamy, 2017*). Peri-sinusoidal Adipo-CAR/MALP progenitor cells are generally primed to undergo adipocyte differentiation; however, they are also recruited to undergo differentiation into trabecular bone osteogenic cells with age (~35 % at 6 months) and into cortical osteoblasts and osteocytes during injury-induced skeletal repair (*Matsushita et al., 2020*; *Baccin et al., 2020*, *Figure 10*). Our results mirror these findings and support an inverse model whereby the depletion of peri-sinusoidal, adiponectin-expressing MALP/Adipo-CAR progenitor cells drives the preferential differentiation of adiponectin[-/lo], Cxcl12[-/lo] progenitors into adipocytes in states of local and

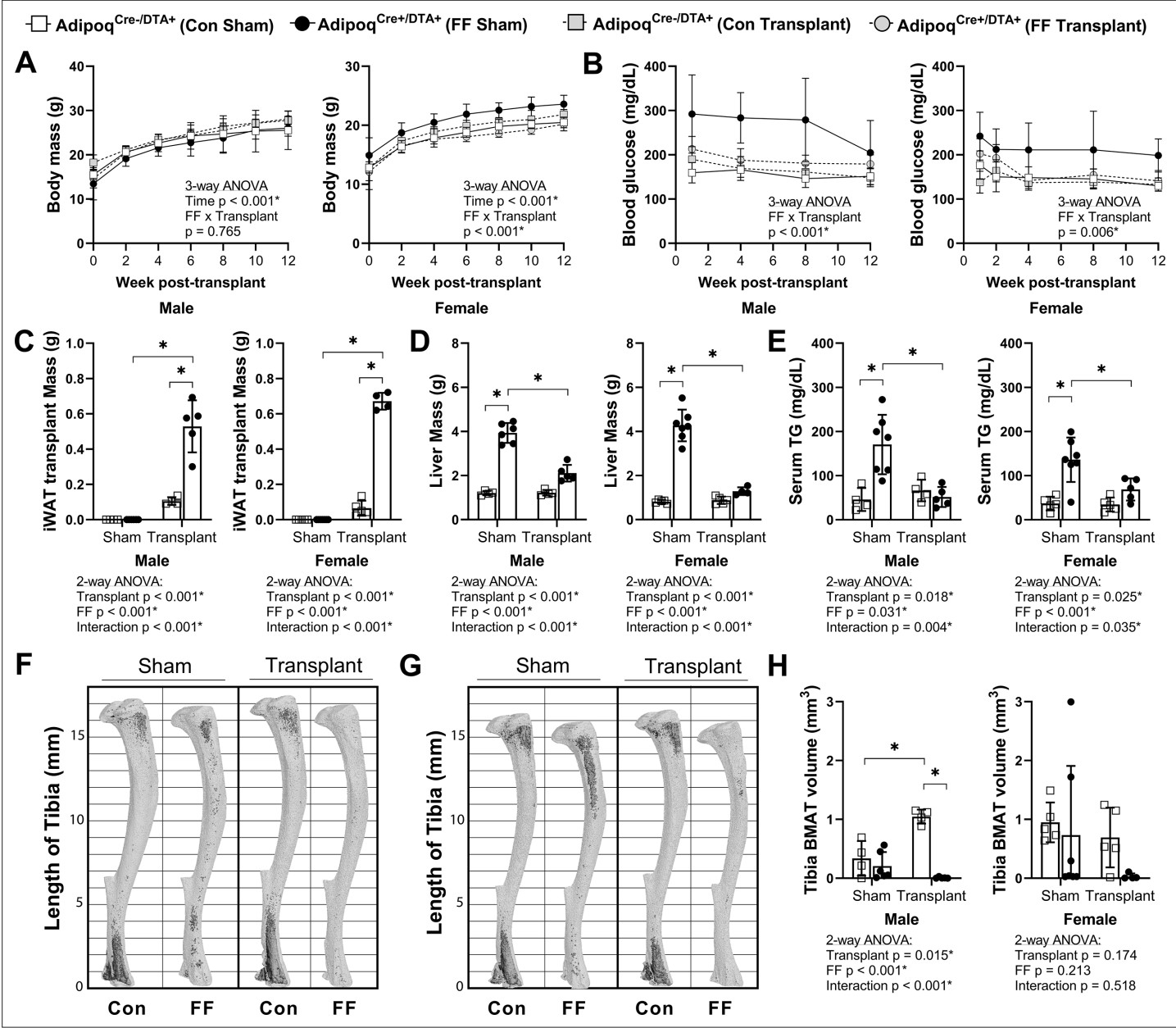

**Figure 8.** Subcutaneous fat transplant prevents BMAT expansion in Adipoq^Cre+/DTA+ fat free (FF) mice. Male and female Adipoq^Cre+/DTA+ FF and Adipoq^Cre-/DTA+ controls (Con) underwent sham surgery or were transplanted subcutaneously with WT inguinal white adipose tissue (iWAT) at 3–5 weeks of age. After surgery, mice were monitored for 12 weeks prior to sacrifice. (**A**) Body mass. (**B**) Random fed blood glucose, measured using a glucometer. (**C**) iWAT transplant mass at endpoint (week 12 after transplantation). (**D**) Liver mass at endpoint. (**E**) Serum triglyceride concentration at 4 weeks after the transplant surgery. (**F, G**) Representative μCT images of osmium-stained tibiae of (**F**) male and (**G**) female mice at endpoint. Bone marrow fat is in dark grey and bone is in light gray. (**H**) Quantification of total tibial BMAT volume. Sample size for control and FF mice, respectively: sham Male n = 4, 6; transplant Male n = 4, 5; sham Female n = 5, 7; transplant Female n = 5, 4. Statistical significance was assessed by (**A, B**) three-way ANOVA and (**C–H**) two-way ANOVA with Tukey's multiple comparisons test. ANOVA results as indicated. *p≤0.05. Data presented as mean ± SD. All mice were housed at 30 °C on a 12 hr/12 hr light/dark cycle.

The online version of this article includes the following source data and figure supplement(s) for figure 8:

**Source data 1.** Fat transplant source data.

**Figure supplement 1.** Engrafted fat transplant fragments resemble subcutaneous white adipose tissue at the terminal end point.

**Figure supplement 2.** Trabecular and cortical bone phenotypes remain unchanged after subcutaneous fat transplant.

**Figure supplement 2—source data 1.** Fat transplant bone parameters.

*Figure 8 continued on next page*

*Figure 8 continued*

**Figure supplement 3.** Adipose tissue masses remain unchanged after subcutaneous fat transplant.

**Figure supplement 3—source data 1.** Fat transplant adipose tissue masses.

systemic metabolic demand, as occurs in the Adipoq^Cre+/DTA+ fat-free mouse model of CGL (*Figure 10*). Because peripheral adipose tissues do not have such an alternate progenitor population within their niche, this may explain why this secondary adipogenesis pathway in FF mice is unique to the bone marrow and is absent in other depots including WAT and BAT. Altogether, this finding contributes to our understanding of the relative preservation of BMAT in clinical states of CGL and reinforces the likely importance of BMAds to maintaining the local homeostasis of the skeletal and hematopoietic microenvironment.

## Adiponectin^-/low, Cxcl12^-/low BMAds have properties of aged BMAT adipocytes and are specialized for lipid storage with reduced expression of adipokines

The ectopic BMAT in FF mice expanded with age and had a larger volume in females than in males, which mirrors results in other models of aging (*Li et al., 2018*). The 87 % decrease in *Cxcl12* expression in FF BMAds also aligns with what has been previously reported in aged BMAds (*Liu et al., 2011*). Specifically, a 46 % decrease in *Cxcl12* expression was observed in BMAds from 18 month old mice relative to BMAds isolated at 6 months (re-analyzed microarray data from *Liu et al., 2011*). Decreased BMAd-specific expression of *Cxcl12* also occurs in obese mice fed with high-fat diet relative to controls (−24 to 41%, re-analyzed microarray data from *Liu et al., 2013*). Decreased expression of *Adipoq* has also previously been highlighted as a feature of aged BMAds (*Liu et al., 2011*). Thus, we propose that expansion of an adiponectin^-/lo, Cxcl12^-/lo BMAd population is a conserved adaptation with age and in states of metabolic stress (*Figure 10*). Functionally, decreases in stromal and BMAd-derived Cxcl12 may lead to decreased focal support of hematopoiesis (*Mattiucci et al., 2018*), contributing to the well-defined pattern of bone marrow atrophy and BMAd expansion that occurs with age and diverse disease states (*Scheller et al., 2016b*; *Scheller and Rosen, 2014a*; *Liu et al., 2011*). In addition to *Cxcl12* and *Adipoq*, FF BMAds had significant decreases in expression of adipokines including adipsin and resistin, suggesting that these cells may have limited endocrine and secretory functions.

Ectopic FF BMAds also had a larger cell size, a deficient response to lipolytic agonists including cold exposure and β3-adrenergic receptor stimulation, and decreased expression of *Adrb3* and *Mgll*. This suggests that FF BMAds are specialized for lipid storage with decreased capacity to serve as a local fuel source for surrounding hematopoietic and osteogenic cells. This result provides insight into recent conflicting studies on lipolysis in rodent and human BMAds (*Scheller et al., 2019*; *Attané et al., 2020*). Specifically, purified BMAds from healthy rodents are capable of responding to lipolytic agonists such as forskolin (*Scheller et al., 2019*). However, purified adipocytes from older humans are not (*Attané et al., 2020*). In humans, this was found to be due to a selective decrease in expression of key lipase *Mgll* (*Attané et al., 2020*), a serine hydrolase that catalyzes the conversion of monoacylglycerides to free fatty acids and glycerol. Similar to aged human BMAds, we found that FF BMAds have decreased expression of *Mgll* with comparable expression of lipases *Lipe* (HSL) and *Pnpla2* (ATGL) relative to controls. This suggests that rodent BMAds can undergo the same adaptations as are present in aged humans, contributing to their resistance to lipolysis. However, unlike humans (*Attané et al., 2020*), we also observed decreased expression of *Adrb3* in mice.

## Improvement of metabolic parameters can prevent ectopic BMAd expansion

Systemic abnormalities including hepatic steatosis, hyperglycemia, and hypertriglyceridemia in FF mice were rescued by subcutaneous fat transplant. Similarly, fat transplant prevented the expansion of ectopic BMAds. This supports the existing paradigm whereby excess circulating lipids contribute to the development and expansion of BMAT, and, conversely, that the decrease of these factors in circulation may reduce the development of ectopic BMAds in bone. It is unclear if this could be accomplished in relatively healthy mice or humans to limit age-associated increases in BMAds in regions of hematopoietic bone marrow and, beyond this, if such a strategy would have benefits to bone or

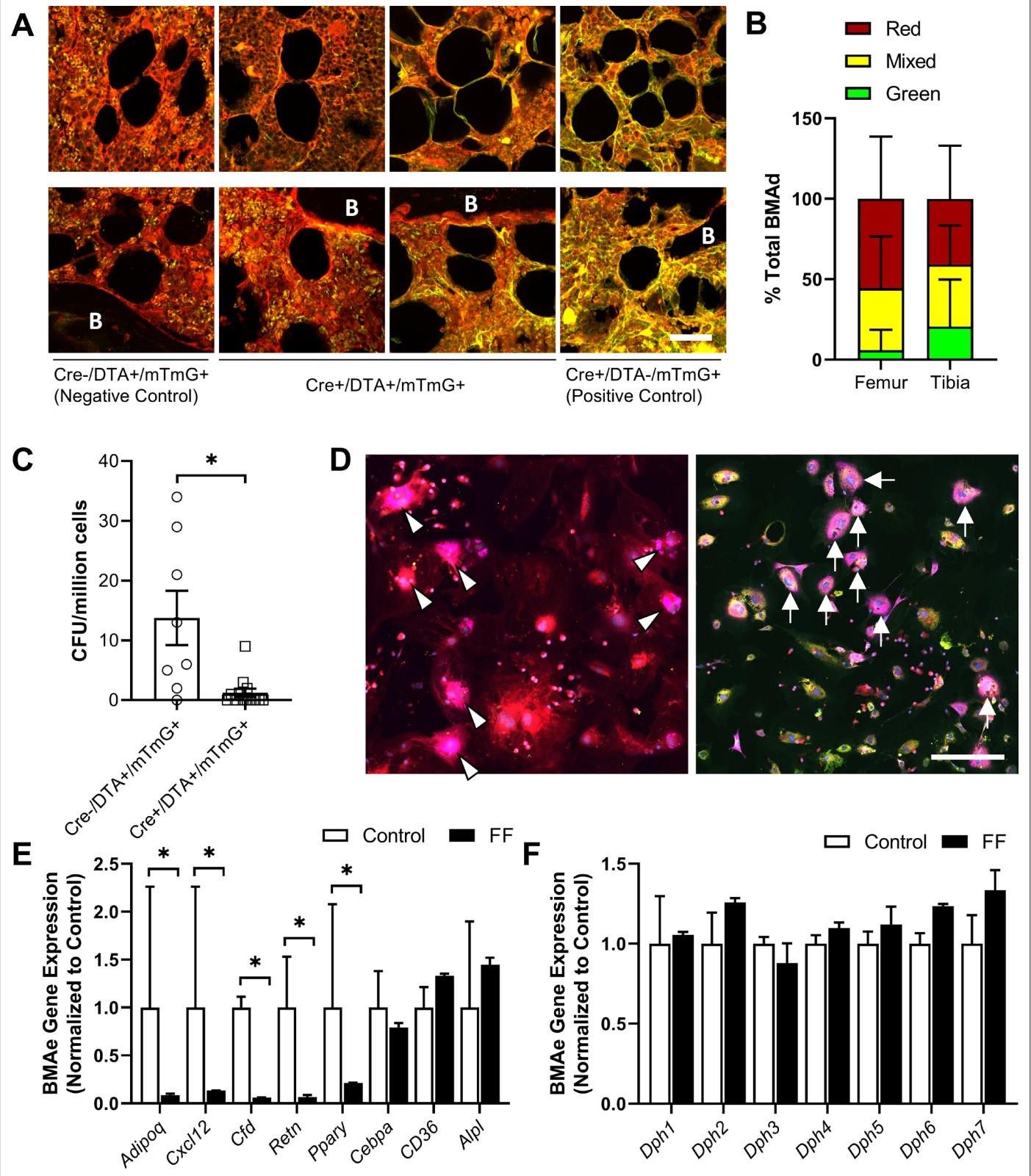

**Figure 9.** Fat-free (FF) bone marrow adipocytes are mostly adiponectin negative and have decreased cytokine expression relative to controls. (**A**) Representative immunostains of bone marrow adipocytes in femur and tibia from both male and female Adipoq^Cre+/DTA+/mTmG+ triple mutant mice (FF^mTmG) at 4 months of age in vivo. Assessed relative to Adipoq^Cre-/DTA+/mTmG+ (negative control) and Adipoq^Cre+/DTA-/mTmG+ (positive control) samples. B = bone. Scale = 50 μm. (**B**) Quantification of GFP+, RFP+, and mixed bone marrow adipocytes (adipocytes verified using PLIN1+ stain, not shown) from

*Figure 9 continued on next page*

*Figure 9 continued*

femur and tibia of FF^mTmG mice. Sample size: femur n = 7; tibia n = 6; total adipocytes counted: n = 802. Unpaired t-test. (**C**) Whole bone marrow from 4-month-old male mice was cultured at low density for 14 days to promote the formation of fibroblastic colony-forming units (CFU-F). Quantification of CFU-F per million bone marrow cells from FF^mTmG or from Adipoq^Cre-/DTA+/mTmG+ control mice (Con^mTmG). Welch's t-test. (**D**) Spontaneous adipogenesis was observed in a subset of residual FF^mTmG stromal cells. Representative adipogenic colonies with GFP+ (white arrows) and RFP+ (white arrowheads) adipocytes demonstrating PLIN1+ lipid droplets (pink). Scale = 200 μm. (**E**) Gene expression of the indicated targets normalized to the geometric mean of housekeeping genes Ppia and Tbp in floated cell preparations enriched for bone marrow adipocytes (BMAe), each gene expressed relative to its respective control. (**F**) Diphthamide biosynthesis protein 1–7 (Dph1–7) gene expression in control and FF BMAe preparations, each gene normalized to its respective control. Control n = 2–4, representative of pooled samples from 20 to 37 mice; FF n = 2, representative of pooled samples from 20 mice. Unpaired t-test with Holm–Sidak correction for multiple comparisons. Data presented as mean ± SD. *p≤0.05. WT and FF mice were housed at 30 °C on a 12 hr/12 hr light/dark cycle.

The online version of this article includes the following source data for figure 9:

**Source data 1.** Percent total BMAd count, CFU per million cells, and BMAe gene expression.

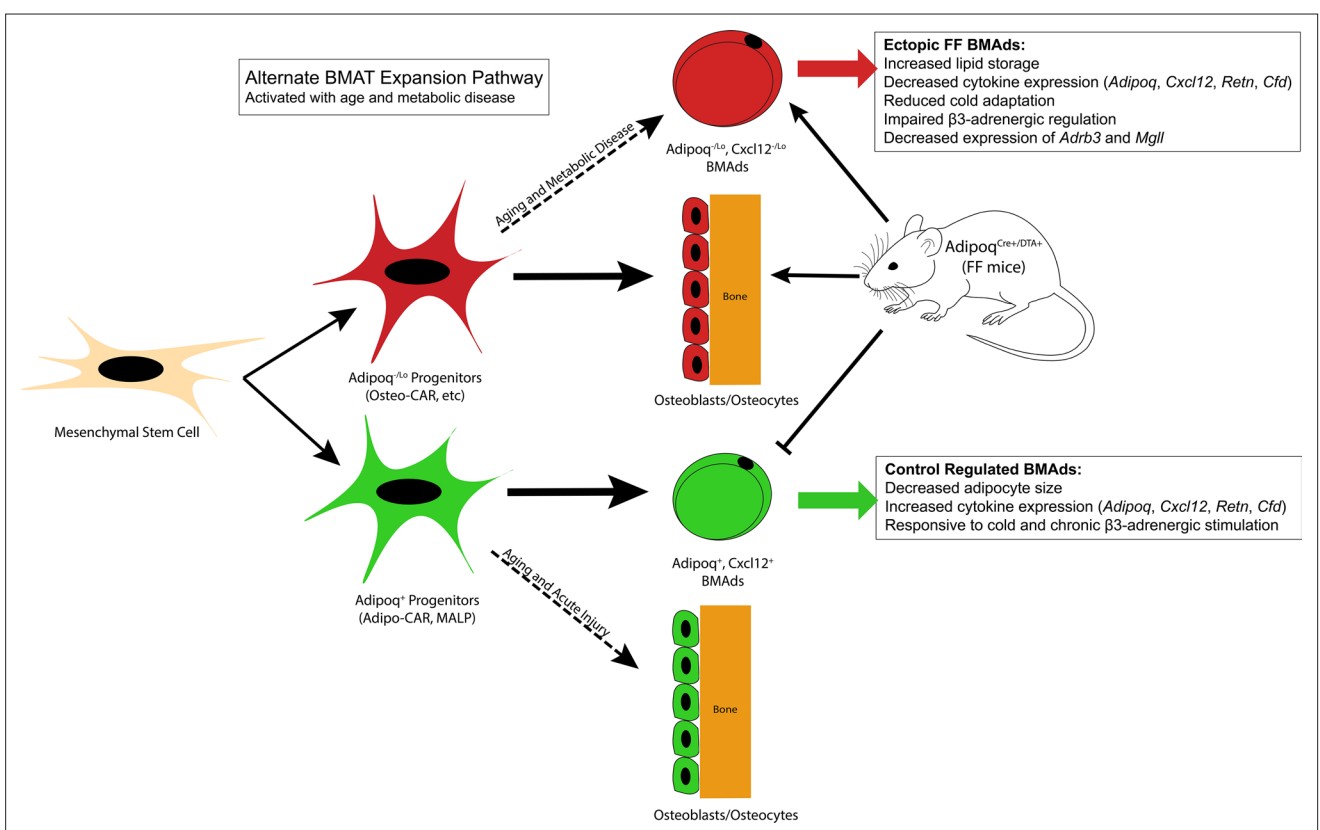

**Figure 10.** Summary model. Adiponectin is highly expressed by mature bone marrow adipocytes (BMAds) and by a subset of bone marrow stromal progenitor cells. Adiponectin-expressing progenitors overlap with Cxcl12-abundant reticular (CAR) cells and have been more recently termed Adipo-CAR cells or MALPs (marrow adipogenic lineage precursors). Adipoq+ skeletal progenitors are primed to undergo adipogenesis. Consistent with this and likely due also to the high expression and secretion of adiponectin by healthy BMAds, classic depots of bone marrow adipose tissue (BMAT) failed to form in the Adipoq^Cre+/DTA+ FF mouse. Instead, we observed age-dependent expansion of a BMAd population with reduced expression of adiponectin (Adipoq^-/lo) and Cxcl12 (Cxcl12^-/lo) in regions of the skeleton such as the diaphysis that are generally devoid of BMAT. FF BMAds were resistant to cold challenge and β3-adrenergic stimulation and had decreased expression of β3-adrenergic receptors and monoacylglycerol lipase (Mgll), suggesting that these cells have decreased capacity to serve as a local fuel source for surrounding hematopoietic and osteogenic cells. We hypothesize that these ectopic BMAds originate from progenitors including the peri-arteriolar Osteo-CAR population, similar to previous work showing that Adipo-CAR cells are capable of undergoing osteogenic differentiation with age and after acute injury. We propose that expansion of this BMAd population is a conserved adaptation with age and in states of metabolic stress and, furthermore, that this is a unique adaptation of the bone marrow that is not present in peripheral adipose depots. Functionally, decreases in stromal and BMAd-derived Cxcl12 may contribute to decreased focal support of hematopoiesis, helping to explain the well-defined pattern of bone marrow atrophy and BMAd expansion that occurs with age and disease.

hematopoiesis. The alteration of other circulating factors such as estradiol, leptin, adiponectin, insulin, corticosterone, growth hormone, and catecholamines may also contribute to BMAT expansion in FF mice. Future work is needed to address these questions.

## Depletion of adiponectin-expressing cells leads to early gains in trabecular bone, but impairs maintenance of cortical bone with age

Specific to bone, μCT scans of the FF tibia revealed a general enhancement of bone formation in young FF mice, as has been reported previously (*Zou et al., 2019*; *Zhong et al., 2020*). However, this was paired with a mild, but significant decrease in tibial length in FF mice, suggesting that the depletion of adiponectin-expressing cells limits bone elongation, potentially through the suppression of chondrocyte differentiation and/or proliferation within the growth plate. This process may also be mediated by reduced levels of growth hormone or growth factors in circulation or within the bone microenvironment. Consistent with this observation, short stature is consistently reported in patients with CGL3, whose BMAT is also preserved (*Kim et al., 2008*). In addition, for the first time, μCT scans of the FF tibia in this study showed a sex- and age-dependent regulation of bone mass after depletion of adiponectin-expressing cells. Specifically, trabecular bone formation in FF mice was more pronounced in females with significant differences in age-related trabecular maintenance. It has previously been reported that these Adipoq$^{Cre+/DTA+}$ FF mice have low levels of estradiol (*Brenot et al., 2020*), which may contribute to the sex-dependent changes in bone that were observed in these mice. The onset of this high bone mass phenotype was not altered by peripheral fat transplant or normalization of circulating glucose/lipid levels, supporting previous work by Zhong et al and suggesting that changes in bone are driven largely by local depletion of adiponectin-expressing cells in the skeleton, independent of peripheral fat loss (*Zhong et al., 2020*). Despite early increases in bone, depletion of adiponectin-expressing cells led to suppression of mineral accrual and a decrease in cortical thickness from 4 to 8 months of age in both male and female FF mice, suggesting that short-term gains may give way to longer-term cortical instability. This supports the previous findings of Matsushita et al. and suggests that adiponectin expressing progenitors, such as MALP/Adipo-CAR cells, may provide necessary contributions to aging or diseased bone through their recruitment to the osteolineage (*Matsushita et al., 2020*).

## Ectopic BMAT is inversely correlated with cortical, but not trabecular, bone mass

Historically, BMAT expansion has been linked to bone loss and osteoporosis (*Fazeli et al., 2013*). In recent years, this assumption has come into question as multiple models have shown that both high bone mass and high BMAT can occur simultaneously (*Scheller et al., 2015*; *Scheller and Rosen, 2014a*; *Fazeli et al., 2013*). In addition, a recent study using *Prx1*-cre mediated knockout of PPARγ demonstrated that BMAT expansion was not necessary for age-associated bone loss in female mice, though it was sufficient to intensify age-dependent cortical porosity (*Almeida et al., 2020*). Though our study was not designed to isolate the direct relationship between bone and BMAT, some observations can be made in the FF model. First, the initial onset of the bone phenotype was independent of the formation of ectopic BMAds, as demonstrated in the fat transplant experiment, suggesting that these early phenotypes are independent. When considered across experiments, trabecular bone volume fraction in the tibia was inversely correlated with metaphyseal BMAT volume in control mice (slope –2.0, $R^2$ 0.137, p=0.037, n = 32). This is consistent with previous reports in aged mice and humans showing that with age, BMAT increases while bone decreases. However, this association was absent in FF mice (slope +0.7, $R^2$ 0.073, p=0.155, n = 29). By contrast, BMAT in the diaphysis was inversely correlated with cortical bone volume fraction in FF mice (slope –2.1, $R^2$ 0.306, p=0.002, n = 29), but not in controls (Slope +1.8, $R^2$ 0.044, p=0.252, n = 32). In healthy mice, the majority of BMAds are derived from Adipoq+ marrow adipocyte lineage progenitors (*Zhong et al., 2020*, *Figure 10*). MALP cells form a peri-sinusoidal network throughout the bone marrow and were recently shown to secrete factors such as RANKL, leading to suppression of trabecular, but not cortical, bone mass (*Yu et al., 2021*). Ablation of these cells in the FF mice may help to explain the absence of a correlation between FF BMAT and trabecular bone. Instead, the ectopic adiponectin$^{-/low}$, Cxcl12$^{-/low}$ adipocyte population appears to be more closely linked to cortical bone. Though we cannot presume causation, we hypothesize that this relationship may be due to the origination of these BMAds from alternate

osteoprogenitor populations such as Osteo-CAR cells that localize to the peripheral arterioles and endocortical surfaces (*Baccin et al., 2020*, *Figure 10*). Future clarification of this point will help to define the niche-specific regulation of bone, blood, and BMAT.

### Limitations

Altogether, these findings suggest that a unique population of BMAds that is specialized for lipid storage with limited lipid mobilization and endocrine functions has the potential to accumulate with age and metabolic disease. This adipogenesis pathway is exclusive to the bone marrow and does not exist in peripheral adipose tissues. However, these results are based on work in the FF model of CGL and have not yet been replicated in alternate rodent models or in humans. Future studies are needed to extend these findings and to explore the implications of this result for skeletal and metabolic health.

### Conclusions

This study demonstrates that the spatially defined progenitor cell, the systemic metabolic profile, and the local microenvironment are necessary regulators of BMAd expansion and adaptation in vivo. In addition, we present evidence for a novel secondary adipogenesis pathway that is unique to the bone marrow and is activated during times of metabolic stress. The resulting adiponectin$^{-/low}$, Cxcl12$^{-/low}$ adipocytes express low levels of adipokines, have increased capacity for lipid storage, and are resistant to lipolytic stimulation, providing new insight into the cellular mechanisms of BMAd adaptation. This work refines our knowledge of the origins of BMAT and contributes to our understanding of the diversity of BMAd populations and their different physiological functions and adaptations within the skeletal system.

# Materials and methods

**Key resources table**

| Reagent type (species) or resource | Designation | Source or reference | Identifiers | Additional information |
|---|---|---|---|---|
| Strain, strain background (*Mus musculus*) | C57BL/6 J: *Adipoq-Cre* | Jackson Laboratories | 028020 RRID:IMSR_JAX:028020 | |
| Strain, strain background (*Mus musculus*) | C57BL/6 J: *Rosa26< lsl-mTmG/+>* | Jackson Laboratories | 007676 RRID:IMSR_JAX:007676 | |
| Strain, strain background (*Mus musculus*) | C57BL/6 J: *Rosa26< lsl-DTA/DTA>* | Jackson Laboratories | 009669 RRID:IMSR_JAX:009669 | |
| Strain, strain background (*Mus musculus*) | C57BL/6 J: *Adipoq$^{-/-}$* | Jackson Laboratories | 008195 RRID:IMSR_JAX:008195 | |
| Chemical compound, drug | 10 % neutral buffered formalin | Fisher Scientific | 23–245684 | |
| Chemical compound, drug | EDTA | Sigma-Aldrich | E5134 | |
| Chemical compound, drug | Hydrogen Peroxide | Sigma-Aldrich | 216,763 | 30 wt. % in $H_2O$ |
| Commercial assay or kit | IMMPRESS HRP Anti-Rabbit IgG kit | Vector Laboratories | MP-7401 RRID:AB_2336529 | |
| Chemical compound, drug | DAPI | Sigma-Aldrich | D9542 | 1 mg/mL |
| Chemical compound, drug | Hematoxylin | Ricca Chemical | 3536–16 | |
| Other | OCT mounting media | Fisher HealthCare | 23-730-571 | |
| Commercial assay or kit | MesenCult Expansion Kit (Mouse) | STEMCELL Technologies | 05513 | |
| Chemical compound, drug | Triton X-100 | Sigma-Aldrich | 9002-93-1 | |
| Other | Fluoromount-G | Thermo Fisher Scientific | 00-4958-02 | |
| Chemical compound, drug | Osmium tetroxide | Electron Microscopy Sciences | 19,170 | 4 % Aqueous Solution |
| Chemical compound, drug | Potassium dichromate | Sigma-Aldrich | 24–4520 | 1/60 M |
| Commercial assay or kit | Serum Triglyceride Determination Kit | Sigma-Aldrich | TR0100 | |
| Chemical compound, drug | CL316,243 | Sigma-Aldrich | C5976 | ≥ 98 % (HPLC) |

*Continued on next page*

*Continued*

| Reagent type (species) or resource | Designation | Source or reference | Identifiers | Additional information |
|---|---|---|---|---|
| Other | Donkey serum | Sigma-Aldrich | D9663 | |
| Other | 4 x NuPage LDS Buffer | Thermo Fisher Scientific | NP0007 | |
| Chemical compound, drug | Ponceau S | Fisher Scientific | BP103-10 | |
| Other | DNase and protease free bovine serum albumin | Fisher Scientific | BP9706 | |
| Other | HBSS Buffer | Gibco | 13150016 | |
| Commercial assay or kit | NucleoSpin RNA XS Kit | Takara Biosciences | 740,902 | |
| Commercial assay or kit | SuperScript IV VILO Master Mix with ezDNase Enzyme | Thermo Fisher Scientific | 11766050 | |
| Commercial assay or kit | qPCRBIO SyGreen Mix Lo-ROX | PCR Biosystems | PB20.11–51 | |
| Antibody | Antibodies for immunostaining and western blot | Detailed in *Supplementary file 1* | | |
| Software, algorithm | GraphPad Prism | GraphPad | v8.4.3 RRID:SCR_002798 | |
| Software, algorithm | SCANCO Medical microCT systems | Scanco Medical AG | RRID:SCR_017119 | |
| Software, algorithm | NDP.view2 | Hamamatsu Photonics | U12388-01 | |
| Software, algorithm | Fiji | ImageJ | RRID:SCR_002285 | |
| Software, algorithm | Microsoft Excel | Microsoft | RRID:SCR_016137 | |
| Other | Scanco μCT 40 | Scanco Medical AG | | Imaging system |
| Other | 2.0-HT NanoZoomer System | Hamamatsu Photonics | | Imaging system |
| Other | Spinning Disk Confocal Microscope | Nikon | | Imaging system |
| Other | LI-COR Odyssey Imager | LI-COR | | Imaging system |
| Other | Glucometer | Contour Next | | |
| Other | Digital Caliper | iKKEGOL | 5486 | |
| Other | PicoLab Rodent Diet 20 | LabDiet | 5053 | |
| Other | Microplate Spectrophotometer | BioTek | Epoch | |
| Other | QuantStudio 3 Real-Time PCR System | Thermo Fisher Scientific A28136 | A28136 | |

## Mice

Institutional guidelines for the handling and experimentation with animals were followed, and all work was approved by the animal use and care committee at Washington University (Saint Louis, MO). All animals were housed on a 12 -hr light/dark cycle and fed ad libitum (PicoLab 5053, LabDiet). Mice were obtained from Jackson Laboratories and bred at Washington University including *Adipoq-Cre* (Strain #028020), *Rosa26-lsl-mTmG* (Strain #007676), *Rosa26-lsl-DTA* (Strain #009669), and *Adipoq* knockout (Strain #008195). For breeding, heterozygous *Adipoq*-Cre+ males were bred to homozygous *Rosa26-lsl-mTmG* or *Rosa26-lsl-DTA* females to generate lineage reporter *Adipoq-Cre, Rosa26-lsl-mTmG* (Adipoq$^{Cre+/mTmG+}$) or *Adipoq-Cre, Rosa26-lsl-DTA* FF (Adipoq$^{Cre+/DTA+}$) mice and associated Adipoq$^{Cre-/mTmG+}$ or Adipoq$^{Cre-/DTA+}$ control littermates. In addition, in a subset of experiments, these mice were further crossed to generate heterozygous triple-mutant Adipoq$^{Cre+/mTmG+/DTA+}$ mice and Adipoq$^{Cre-/mTmG+/DTA+}$ controls. Mice expressing DTA under the control of the *Adipoq-Cre* promoter lack both white and brown adipose tissues and were bred and housed at thermoneutral temperature (30 °C). All transgenic mice were maintained on a C57BL/6 J background (Strain #000664). Body mass was recorded with an electronic scale, and blood glucose was monitored by tail prick with a glucometer (Contour Next). For end points requiring tissue mass measurements, mice were euthanized with carbon dioxide followed by cervical dislocation. Tissues were collected and weighed using an electronic scale. For end points requiring histology and immunostaining, mice were anesthetized with ketamine/xylazine cocktail (100 mg/kg ketamine; 10 mg/kg xyalzine) and perfused through the left ventricle of the heart with 10 ml phosphate-buffered saline followed by 10 ml 10 % neutral buffered

formalin (NBF; Fisher Scientific 23–245684). When indicated, tibia and femur lengths were determined using a digital caliper (iKKEGOL). For all experiments, collected tissues were post-fixed in 10 % NBF for 24 hr. For western blot and serum assays, as detailed below, blood was collected through capillary action from the lateral tail vein, and serum was isolated by centrifugation at 1500 × g for 15 min after clotting on ice.

## Western blot

Immunoblotting for serum adiponectin (constant volume) was performed as described previously (*Cawthorn et al., 2014*). Specifically, serum samples were reduced and denatured in 4× NuPage LDS sample buffer (ThermoFisher, NP0007) containing 1:8 parts β-mercaptoethanol (2 µl serum +10 µl LDS buffer +28 µl water). Preparations were incubated at 95 °C for 5 min and cooled on ice for 1 min before separating by SDS–PAGE. After transfer to PVDF membrane, HRP-conjugated secondary antibody to adiponectin (*Supplementary file 1*) was visualized with Western Lightning Plus (Perkin Elmer, Waltham, MA) and imaged using a LI-COR Odyssey Imager (LI-COR Biosciences, Lincoln, NE). After immunoblotting, the membrane was stained for 1 min with Ponceau S as a loading control (0.5% w/v in 1 % acetic acid, Fisher, BP103-10). Ponceau-stained membranes were rinsed with water prior to drying and imaging.

## Histology and immunostaining

### Paraffin immunostaining and imaging

Paraffin embedding, slide preparation, and H&E stains were performed by the WUSM Musculoskeletal Histology and Morphometry core. Bones were fully decalcified in 14 % EDTA (Sigma-Aldrich E5134), pH 7.4 prior to embedding. For immunostaining, 10 µm paraffin sections were rehydrated in a series of xylene and ethanols prior to antigen retrieval with 10 mM sodium citrate buffer (pH 6.0, 20 min, 90–95°C or overnight at 55 °C). Antibodies used for paraffin immunostaining are detailed in *Supplementary file 1*. Paraffin Immunofluorescence: Retrieved sections were permeabilized for 10 min in 0.2 % Triton-X in PBS, blocked for 1 hr at room temperature with 10 % donkey serum (Sigma-Aldrich D9663) in TNT buffer (0.1 M Tris–HCL pH 7.4, 0.15 M sodium chloride, 0.05 % Tween-20), and incubated for 24 hr at 4 °C with primary antibodies followed by washing and secondary detection (*Supplementary file 1*). Secondary antibodies in TNT buffer were applied for 1 hr at room temperature. Nuclei were counterstained in 1 µg/ml DAPI (Sigma-Aldrich D9542) for 5 min prior to mounting in Fluoromount-G (ThermoFisher, 00-4958-02). All washes between steps were performed three times each in TNT buffer. Paraffin Immunohistochemistry: Tissue sections were permeabilized for 10 min in 0.2 % Triton-X in PBS, blocked for 1 hr in kit-specific blocking reagent (ImmPRESS HRP Goat Anti-Rabbit IgG Polymer Detection Kit, Vector Laboratories, MP-7451), and incubated for 24 hr at 4 °C with primary antibody (*Supplementary file 1*). Sections were washed in TNT, and endogenous peroxidase activity was quenched in 0.3 % hydrogen peroxide (Sigma-Aldrich 216763) in PBS for 30 min. Sections were then incubated with ImmPRESS polymer reagent for 30 min prior to development with peroxidase substrate solution. Slides were counterstained with hematoxylin (Ricca Chemical 3536–16) and dehydrated through a reverse ethanol gradient prior to mounting in Permount. Images were taken using a Nikon Spinning Disk confocal microscope or a Hamamatsu 2.0-HT NanoZoomer System with NDP.scan 2.5 image software.

### Frozen immunostaining and imaging

Tissues were embedded in OCT mounting media (Fisher HealthCare 23-730-571) and cut at 50 µm on a cryostat (Leica). Bones were fully decalcified in 14 % EDTA, pH 7.4 prior to embedding. Sections were blocked in 10 % donkey serum in TNT buffer prior to incubation for 48 hr with primary antibodies at 4 °C (*Supplementary file 1*). After washing, secondary antibodies in TNT buffer were applied for 24  hr at 4 °C (*Supplementary file 1*). The sections were then washed and incubated in DAPI for 5 min prior to mounting with Fluoromount-G. Images were taken at 10 × on a Nikon spinning disk confocal microscope.

## BMSC isolation, cell culture, and immunostaining

Immediately after euthanasia by $CO_2$, long bones were harvested under aseptic conditions. The ends of the long bones were cut to allow flushing of marrow contents, as described previously (*Scheller*

*et al., 2010*). Cells were suspended in MesenCult Expansion Medium (STEMCELL Technologies 05513) containing MesenCult Basal Medium, MesenCult 1 × Supplement, 0.5 ml MesenPure, 1 × L-glutamine, and 1 × penicillin/streptomycin. Primary bone marrow cultures were plated at a density of $1 \times 10^5$ cells/cm$^2$ in tissue culture-treated imaging plates (for immunostaining) or six-well plates (for CFU-F assay) and incubated at 37 °C, 5 % $CO_2$. After 48 hr, non-adherent cells were removed with subsequent media changes occurring every 2–3 days. After 14 days, colonies were fixed with methanol prior to staining with crystal violet (for CFU-F assay) or permeabilization with 1 % Triton X-100 (Sigma-Aldrich 9002-93-1) in PBS for 10 min at room temperature (for immunostaining). Cells were blocked with a solution containing PBS, 10 % donkey serum, and 0.1 % Triton X-100 for 30 min prior to incubation for 24 hr at 4 °C with primary antibodies (*Supplementary file 1*). Cells were then washed prior to application of secondary antibodies in PBS and 0.1 % Triton X-100 for 30 min at room temperature. Nuclei were stained with 1 µg/ml DAPI for 5 min prior to mounting in Fluoromount-G. Images were taken at 4 × and 20 × using a Nikon spinning disk confocal microscope.

## Computed tomography and osmium staining

Bones were embedded in 2 % agarose prior to scanning at 20 µm voxel resolution using a Scanco µCT 40 (Scanco Medical AG). Analysis was performed according to reported guidelines (*Bouxsein et al., 2010*). For cancellous bone, 100 slices (2 mm) below the growth plate, beginning where the primary spongiosa was no longer visible, were contoured and analyzed at a threshold of 175 (on a 0–1000 scale relative to a pre-calibrated hydroxyapatite phantom). For cortical bone, 20 slices (400 µm) located 2 mm proximal to the tibia–fibula junction were contoured and analyzed at a threshold of 260. To assess bone marrow adiposity, bones were decalcified in 14 % EDTA, pH 7.4 and incubated in a solution containing 1 % osmium tetroxide (Electron Microscopy Sciences 19170) and 2.5 % potassium dichromate (Sigma-Aldrich 24–4520) for 48 hr (*Scheller et al., 2014b*). After washing for 2 hr in running water and storage in PBS at 4 °C, osmium-stained bones were embedded in 2 % agarose and scanned at 10 µm voxel resolution (Scanco µCT 40; 70 kVp, 114 µA, 300 ms integration time). Regions of interest were contoured for BMAT quantification as detailed in the figure legends. BMAT was segmented with a threshold of 400.

## BMAd cell size analysis

Tiled 10 × images covering the femoral and tibial metaphyses were exported from the Nanozoomer scans of H&E stained slides and processed in Fiji to estimate average adipocyte cell size (*Schindelin et al., 2012*). Based on previous recommendations for adipocyte cell size analyses (*Parlee et al., 2014*), a minimum of 100 adipocytes were analyzed for each mouse. Briefly, the scale in Fiji was set to be consistent with the original scan. The image was then converted to 8-bit and a threshold of 230–255 was applied to create a mask. Then the image was cleaned up using the wand tool and the deletion command to eliminate non-adipocyte structures. The cleaned mask was processed using the Fill Holes and the Watershed tools. The size of adipocytes was determined using the 'Analyze Particles' tool by setting the size to 200–4000 µm$^2$ and circularity to 0.40–1.00. Histograms were created in GraphPad Prism, and the average adipocyte cell size was calculated using Excel.

## CL316,243 injection

CL316,243 (Sigma-Aldrich, C5976) was reconstituted in saline to a concentration of 0.01 mg/ml and stored at 4 °C for up to 2 weeks. Eight daily subcutaneous injections of 0.03 mg/kg CL316,243 were administered over the course of 10 days (weekdays only, M→F, M→W) prior to sacrifice on Day 11.

## BMAd purification, RNA extraction, and qPCR

BMAds were collected from groups of 8–12 mice at 4–6 months of age. Femurs and tibiae (16–24 bones/preparation) were rapidly dissected into pre-warmed 37 °C HBSS buffer (Gibco 10425–076) containing 2 % DNase and protease-free bovine serum albumin (Fisher BP9706), 5 mM EDTA, and 1 g/l glucose. After cutting the ends of the bones, whole bone marrow was flushed into a 50 ml conical tube with a 10 ml syringe +22 gauge needle and resuspended into 20 ml fresh buffer +1 mg/ml collagenase. Marrow-depleted bones were placed into a separate tube in 20 ml buffer +1 mg/ml collagenase and finely minced to liberate any residual BMAds. Bone and bone marrow preparations were

centrifuged at room temperature, 400 g × 2 min, and BMAd-containing supernatant was decanted into a new tube prior to re-centrifugation at 400 g × 1 min. Infranatant and any residual pellet were removed using a pulled glass pipet until only 1–2 ml of liquid was remaining. The adipocyte-containing liquid was serially applied to a NucleoSpin Filter Column (NucleoSpin RNA XS Kit, Takara, 740902) for on-column BMAd lysis and RNA extraction. Briefly, the filter column was centrifuged slowly at 50 g × 10 s to retain the BMAd cells while removing any residual liquid into the collection tube. The bottom of the column was then sealed with parafilm, and kit-supplied RNA lysis buffer was added with gentle agitation. BMA-enriched ('BMAe') lysates were processed for RNA extraction using the kit-supplied protocol and reagents.

For qPCR, 100 ng of total RNA was reverse transcribed into cDNA using SuperScript IV VILO Master Mix with ezDNase Enzyme (Thermo Fisher Scientific 11766050) according to the manufacturer's instruction. SyGreen 2 × Mix Lo-ROX (PCR Biosystems PB20.11–51) was used to perform the qPCR assay on a QuantStudio 3 Real-Time PCR System (Thermo Fisher Scientific A28136). Gene expression of individual targets was calculated based on amplification of a standard curve for each primer. Results were normalized to the geometric mean of housekeeping genes *Ppia* and *Tbp*. Primer sequences are listed in *Supplementary file 2*.

## Fat transplantation

*Adipoq*[Cre-/DTA+] and *Adipoq*[Cre+/DTA+] mice received subcutaneous fat transplant or sham surgery at 3–5 weeks of age. Mice were maintained for an additional 12 weeks prior to sacrifice (end age 15–17 weeks). Donor preparation: wild-type donor mice on the same background (C57BL/6 J) ranged from 22 to 39 days of age. Immediately after decapitation under anesthesia, bilateral inguinal WAT depots were dissected free of surrounding tissues and placed into sterile PBS in a petri dish. The lymph node was removed, and a scalpel blade was used to mince the remaining iWAT into small pieces of ~0.5–1.0 mm³. The entire minced iWAT from one donor mouse was transplanted to one recipient mouse. Recipient surgery: the recipient mouse was anesthetized with isoflurane and the skin on the back was prepared (shaved and treated 2 × each with 70 % ethanol and betadine) prior to making two 1 cm incisions along the midline, one over the shoulder blades, and one just above the level of the pelvis. Blunt dissection was used to create four pockets just lateral to each incision, one on each side. The minced iWAT from the donor mouse was evenly distributed into the four pockets. The incisions were closed, and all mice received Buprenex SR at the time of surgery for post-operative analgesia. Post-surgical monitoring and management were performed per DCM guidelines, as approved in our animal protocol.

## Serum glycerol and triglyceride assay

Serum glycerol and true triglyceride (TG) levels were determined using a Serum Triglyceride Determination Kit (Sigma-Aldrich TR0100). In brief, free glycerol reagent and triglyceride reagent were prepared according to the manufacturer's instruction. To measure serum glycerol, 10 µl serum/well was added to a 96-well microplate on ice prior to addition of 150 µl of free glycerol reagent and incubation at 37 °C for 10 min. The absorbances of the standards and the samples at 540 nm versus blank (pure Free Glycerol Reagent) were measured using a microplate spectrophotometer (BioTek). To determine serum true TG level, 38 µl of triglyceride reagent was added to each well after the initial absorbance measurement for glycerol, followed by an additional 10 min incubation at 37 °C. The absorbances of the standards and the samples at 540 nm versus blank were measured again using the microplate reader. A standard curve was utilized for the calculation of serum-free glycerol and total TG concentrations. The serum true TG level was calculated by subtracting the free glycerol level from the total TG level for each sample, as per manufacturer instructions. All samples were assayed in duplicate.

## Statistics

Statistical analyses were performed in GraphPad Prism including unpaired t-test, one-way, two-way, and three-way ANOVA with multiple comparisons tests, applied as detailed in the figure legends. A p-value of less than 0.05 was considered statistically significant. Quantitative assessments of cell size and µCT-based analyses were performed by individuals that were blinded to the sample identity.

## Acknowledgements

This work was supported by grants from the National Institutes of Health including R00-DE02417 and startup funds from the Washington University Department of Medicine. We are also grateful for the core services provided by the Musculoskeletal Research Center (NIH P30-AR074992) and the Washington University Center for Cellular Imaging (supported by the Washington University School of Medicine, The Children's Discovery Institute of Washington University, and St. Louis Children's Hospital CDI-CORE-2015–505 and CDI-CORE-2019–813 and the Foundation for Barnes-Jewish Hospital 3770 and 4642). Lastly, we would like to extend special thanks to Dr. Jesse Procknow for technical assistance and to Dr. Steven Teitelbaum and Dr. Wei Zou for their helpful discussions during the initial stages of this project.

## Additional information

### Funding

| Funder | Grant reference number | Author |
| --- | --- | --- |
| National Institutes of Health | R00-DE02417 | Erica L Scheller |

The funders had no role in study design, data collection and interpretation, or the decision to submit the work for publication.

### Author contributions

Xiao Zhang, Conceptualization, Data curation, Formal analysis, Investigation, Validation, Visualization, Writing – original draft, Writing – review and editing; Hero Robles, Data curation, Formal analysis, Investigation, Methodology, Validation, Visualization, Writing – original draft, Writing – review and editing; Kristann L Magee, Madelyn R Lorenz, Zhaohua Wang, Data curation, Investigation, Writing – review and editing; Charles A Harris, Conceptualization, Resources, Supervision, Writing – review and editing; Clarissa S Craft, Data curation, Formal analysis, Funding acquisition, Investigation, Methodology, Project administration, Validation, Visualization, Writing – review and editing; Erica L Scheller, Conceptualization, Data curation, Formal analysis, Funding acquisition, Investigation, Methodology, Project administration, Resources, Supervision, Validation, Visualization, Writing – original draft, Writing – review and editing

### Author ORCIDs

Xiao Zhang http://orcid.org/0000-0003-2427-0147
Hero Robles http://orcid.org/0000-0002-6439-1309
Erica L Scheller http://orcid.org/0000-0002-1551-3816

### Ethics

All work was performed as approved by the Institutional Animal Care and Use Committee (IACUC) at Washington University (Saint Louis, MO, USA; Protocol IDs 20160183 and 20180282). Animal facilities at Washington University meet federal, state, and local guidelines for laboratory animal care and are accredited by the Association for the Assessment and Accreditation of Laboratory Animal Care (AAALAC).

### Decision letter and Author response

Decision letter https://doi.org/10.7554/eLife.66275.sa1
Author response https://doi.org/10.7554/eLife.66275.sa2

## Additional files

### Supplementary files
- Supplementary file 1. Antibodies used for western blot and immunostaining.
- Supplementary file 2. qPCR Primers.

• Transparent reporting form

## Data availability

All data generated or analyzed during this study are included. Supporting files, including source data for all figures, are available as part of the published article. Reagent information, antibody use details, and primer sequences are provided in the Key resources table and Supplementary files 1 and 2.

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
