## [Decision Letter]

**Acceptance summary:**

The reviewers found the study of bone marrow adipocytes in fat-free mice quite interesting. They also acknowledge that the manuscript comes from a prominent group with long-standing interest in bone marrow adipose tissue and its characteristics relatively to the white and brown adipose tissues. The identification of bone-specific adipogenic pathway, and characterization of ectopic bone marrow adipocytes in aging and under metabolic stress were considered major strengths of the study.

**Decision letter after peer review:**

Thank you for submitting your article "A bone-specific adipogenesis pathway defines key origins and adaptations of bone marrow adipocytes with age and disease" for consideration by *eLife*. Your article has been reviewed by 3 peer reviewers, and the evaluation has been overseen by a Reviewing Editor and Carlos Isales as the Senior Editor. The reviewers have opted to remain anonymous.

Essential revisions:

– The reviewers find that this study represents a continuation of the work of the Scheller group using a previously published mouse model with unique characteristics of bone marrow adipocytes. Moreover, the use of this artificial fat-free mouse model was seen as a limitation of the study, as the bone-specific adipogenesis pathway may in fact be relevant only to the context of adiponectin-free fat-free mice. In line of this, it is recommended that the authors establish more cearly the novelty of the study and the specific knowledge and impact it is contributing.

– The use of the fat-free mouse model is not fully justified. The authors do not experimentally show the efficacy of DTA-mediated cell ablation, which is notoriously ineffective. They should specifically address the possibility that bone marrow adipocytes in Adipoq-Cre/DTA mice are those that escaped from recombination.

– More specifically, the reviewers questioned if Figure 1 and much of Figure 2 are documenting novel information, as they appear very close to the previously published work of the same group (Robles et al., 2019). Also, Figures 3 and 4 provide evidence that fat free mice have a high bone mass phenotype, in support of well-established findings for lipodystrophy in humans, and high bone mass in fat-free mice, and therefore appear redundant. The authors would need to explain the novelty and utility of these data, and their relation to the state of the art in the field.

– Figure 6 was found to be of most interest, as it presents new and exciting data that the global deletion of adiponectin-expressing adipocytes did not affect the presence of bone marrow adipocytes and that this pool of adipocytes even expands with age. These observations are novel and warrant mechanistic analysis. However, the authors present only 5 genes that may be differentially regulated, and without clear interpretation of the experimental findings (Figure 6F). The authors explain how and why the bone marrow adipocytes differ from other adipocytes. Also, it would be important to clarify if the cell deletion occurred equally well in all populations of adipocytes.

– The secondary adipogenesis pathway may in fact be relevant only in extreme circumstances such as in congenital generalized lipodystrophy. Also, the "bone-specific adipogenesis pathway" has been documented only for adiponectin-free fat-free mice. The authors should therefore add "in fat-free mice" at the end of the title, and revise the abstract which appears to be oversimplified and overstated and the conclusions some of which are not supported by experimental evidence.

– Similarly, the stated hypothesis is not really a hypothesis, because this point has been demonstrated in a number of previous studies. The real novelty is that adiponectin-negative bone marrow adipocytes appear under special and rather extreme circumstances. It would be essential to demonstrate that these bone marrow adipocytes in FF mice are indeed adiponectin-negative (by immunostaining).

– The authors do not consider the possibility that DTA does not kill all adiponectin-expressing cells. The DTA allele is known to be ineffective and insensitive to Cre recombinations. This is important as leakiness could undermine the major conclusions of this study. The authors should therefore determine the efficiency of cell ablation, by combining DTA and mTmG alleles with Adipoq-Cre. Any adiponectin-expressing cell escaping DTA should turn GFP+.

---

## [Author Response]

Essential revisions:– The reviewers find that this study represents a continuation of the work of the Scheller group using a previously published mouse model with unique characteristics of bone marrow adipocytes. Moreover, the use of this artificial fat-free mouse model was seen as a limitation of the study, as the bone-specific adipogenesis pathway may in fact be relevant only to the context of adiponectin-free fat-free mice. In line of this, it is recommended that the authors establish more cearly the novelty of the study and the specific knowledge and impact it is contributing.

Thank you for the comments. We have updated the title and abstract to more clearly state that this work was completed in the fat-free model. As requested, we have also revised the Discussion section to more clearly highlight the novelty, specific knowledge, and impact of the study. Specifically, we have added concise Discussion section headings as indicated below. In addition, we have added a limitations section to address the use of the fat-free model. Revisions to the main text are listed below.

Revised title:

“A bone-specific adipogenesis pathway in fat-free mice defines key origins and adaptations of bone marrow adipocytes with age and disease”

Revised statement in abstract:

“This pathway is unique to the bone marrow and is activated with age and in states of metabolic stress in the fat-free mouse model, resulting in the expansion of bone marrow adipocytes specialized for lipid storage with compromised lipid mobilization and cytokine expression within regions traditionally devoted to hematopoiesis.”

New Discussion section headings:

– Ectopic FF BMAds originate from adiponectin-/low progenitors and express low levels of Cxcl12

– Adiponectin-/low, Cxcl12-/low BMAds have properties of aged BMAT adipocytes and are specialized for lipid storage with reduced expression of adipokines

– Improvement of metabolic parameters can prevent ectopic BMAd expansion

– Depletion of adiponectin-expressing cells leads to early gains in trabecular bone, but impairs maintenance of cortical bone with age

–Ectopic BMAT is inversely correlated with cortical, but not trabecular, bone mass

New limitations section:

“Limitations

Altogether, these findings suggest that a unique population of BMAds that is specialized for lipid storage with limited lipid mobilization and endocrine functions has the potential to accumulate with age and metabolic disease. This adipogenesis pathway is exclusive to the bone marrow and does not exist in peripheral adipose tissues. However, these results are based on work in the FF model of CGL and have not yet been replicated in alternate rodent models or in humans. Future studies are needed to extend these findings and to explore the implications of this result for skeletal and metabolic health.”

– The use of the fat-free mouse model is not fully justified. The authors do not experimentally show the efficacy of DTA-mediated cell ablation, which is notoriously ineffective. They should specifically address the possibility that bone marrow adipocytes in Adipoq-Cre/DTA mice are those that escaped from recombination.

Thank you for this important suggestion. To explore the origins of this unique BMAd population we created and analyzed triple mutant fat-free *Adipoq*^Cre+/DTA+/mTmG+^ lineage tracing reporter mice (FF^mTmG^). This work is now presented in a new section of the results entitled “Origins and adaptations of ectopic BMAT adipocytes” with a new main figure (appended below). We have also inserted the description of this triple mutant mouse model into the Introduction and Methods sections and added additional interpretation of these data to the Discussion section.

New Results, Figure, and Figure Legend:

“Origins and adaptations of ectopic BMAT adipocytes

To further determine the origin of ectopic BMAds in FF mice while testing the efficacy of the DTA, we generated triple mutant Adipoq^Cre+/DTA+/mTmG+^ lineage tracing reporter mice (FF^mTmG^). Adipoq^Cre-/DTA+/mTmG+^ mice were used as negative controls (Con^mTmG^). In the FF^mTmG^ model, *Adipoq*-expressing cells will express GFP and should undergo DTA-mediated cell death. By contrast, cells that do not express *Adipoq* will express RFP. To begin, cross-sections of femur and tibia from male and female FF^mTmG^ at 4-months of age were imaged after immunostaining for GFP, RFP, and PLIN1 to assess adiponectin expression in the ectopic BMAds. Unexpectedly, both adiponectin-negative (RFP+, 41-56%) and adiponectin-expressing (GFP+, 6-21%) BMAT adipocytes were present in FF^mTmG^ mice (Figure 9A,B). In addition, approximately 38% of cells had indeterminate membranes, with evidence of both RFP and GFP expression in vivo (Figure 9A,B). Next, we plated primary BMSCs from Con^mTmG^ and FF^mTmG^ mice for quantitative CFU assays (Figure 9C). Consistent with the broad expression of *Adipoq* in the stromal compartment (Figure 2,S1), the number of CFUs was reduced by 91% in FF^mTmG^ mice (Figure 9D). In addition, in line with our in vivo results, spontaneous adipogenesis in the residual FF^mTmG^ BMSCs was observed in both adiponectin-negative (RFP+) and adiponectin-expressing (GFP+) stromal progenitor cells in vitro (Figure 9C)*.* This result suggests that the majority of FF BMAds originate from adiponectin-negative stromal cells, with a minor portion of BMAds being adiponectin-positive. This minor population of cells was not sufficient to restore circulating adiponectin, as adiponectin levels in FF mice were not increased when compared to adiponectin knockout animals (Figure 2A).

Next, we analyzed the gene expression of purified BMAds from control and FF mice. As expected, expression of *Adipoq* was decreased in BMAds from FF mice (Figure 9E). In addition, expression of cytokines including stromal cell-derived factor 1, also known as C-X-C motif chemokine 12 (*Cxcl12*), adipsin (*Cfd*), and resistin (*Retn*) were significantly decreased (Figure 9E). Expression of adipogenic transcription factor peroxisome proliferator-activated receptor γ (*Ppary*) was also decreased. By contrast, expression of CCAAT/enhancer-binding protein α (*Cebpa*), fatty acid transporter *Cd36*, and alkaline phosphatase (*Alpl*) were comparable in control and FF BMAds. Lastly, we assessed the expression of diphthamide biosynthesis enzymes 1-7 (*Dph1-7*), as deficiency in even a single *Dph* enzyme can confer resistance to DTA-mediated cell death (1,2). We did not observe significant regulation of any *Dph* genes in FF BMAds relative to control BMAds (Figure 9F). Overall, our findings define the FF BMAd as an ectopically positioned, PLIN1+, CD68- adipocyte that is specialized for lipid storage with decreased capacity for lipid mobilization and expression of cytokines including adiponectin, resistin, adipsin, and *Cxcl12*.”

– More specifically, the reviewers questioned if Figure 1 and much of Figure 2 are documenting novel information, as they appear very close to the previously published work of the same group (Robles et al., 2019). Also, Figures 3 and 4 provide evidence that fat free mice have a high bone mass phenotype, in support of well-established findings for lipodystrophy in humans, and high bone mass in fat-free mice, and therefore appear redundant. The authors would need to explain the novelty and utility of these data, and their relation to the state of the art in the field.

Thank you for these notes. Yes, partial in vivo data from the *Adipoq*^Cre+/mTmG+^ reporter mice in Figure 1 was previously reported in another study from our group (Craft et al., 2019; PMID: 31758074). However, in the current study, we are providing new images and additional details regarding tracing to bone cells. We think that these data provide important context for the current in vivo adiponectin DTA model. These results also inform the ongoing controversy regarding the expression of adiponectin by skeletal lineage cells. Specifically, our results indicate that adiponectin-expressing progenitors do not give rise to embedding osteoblasts in relatively young, healthy mice, as we found that though the adiponectin lineage reporter traces to many bone lining cells, it is not expressed in the osteocytes. We believe that this is an important point that supports the work by Ono group (Matsushita et al., 2020) as they showed that adiponectin-expressing stromal cells are a committed pre-adipocyte population in healthy conditions but can form osteoblasts/osteocytes only with injury or advanced age. In response to the reviewers, we have now moved these data into the supplement (Figure 1—figure supplement 1; this style of figure nomenclature was requested by the *eLife* editorial team).

In Figure 2, the in vitro culture of the bone marrow stromal cells from the *Adipoq*^Cre+/mTmG+^ reporter mice is a new, unpublished set of data that further supports the in vivo data and shows that adiponectin is expressed in a subset of bone marrow stromal cells. Through this new in vitro study, we confirm that these cells function as pre-adipocytes, with adiponectin being expressed during progenitor commitment but prior to lipid accumulation in healthy cells. This provides additional support for this emerging paradigm that has been recently established by the Qin group (Zhong et al., 2020) and the Ono group (Matsushita et al., 2020). We request to keep this Figure and new data in the main text (Figure 1 in the revised draft). In response to the reviewers, we have also added an additional statement in the 1^st^ paragraph of the discussion to describe the novelty and utility of the in vitro lineage tracing data:

“Ex vivo culture of primary bone marrow stromal cells from *Adipoq*^Cre+/mTmG+^ reporter mice in this study further demonstrated their ability to differentiate into mature adipocytes (Figure 1), supporting these previous findings.”

Regarding the novelty of the bone data in Figure 4 and Supplemental S1, in this study we analyzed both male and female mice at both 4- and 8-months of age. Previous reports have focused on male mice only and have not explored age-related effects. We identified sex-specific differences in cortical and trabecular bone phenotypes, which have never been reported before. We also found, for the first time, that there are key issues in the maintenance of cortical bone with age after depletion of adiponectin-expressing cells and that ectopic BMAd expansion is inversely correlated with cortical, but not trabecular, bone mass. Given the importance and novelty of these results, we would request to keep the figures as part of the manuscript (current Figure 3 and Figure 3—figure supplement 1). In response to the reviewers, we have updated the text of the bone-related Discussion sections to more clearly emphasize these results under the following two headings:

– Depletion of adiponectin-expressing cells leads to early gains in trabecular bone, but impairs maintenance of cortical bone with age

– Ectopic BMAT is inversely correlated with cortical, but not trabecular, bone mass

– Figure 6 was found to be of most interest, as it presents new and exciting data that the global deletion of adiponectin-expressing adipocytes did not affect the presence of bone marrow adipocytes and that this pool of adipocytes even expands with age. These observations are novel and warrant mechanistic analysis. However, the authors present only 5 genes that may be differentially regulated, and without clear interpretation of the experimental findings (Figure 6F). The authors explain how and why the bone marrow adipocytes differ from other adipocytes. Also, it would be important to clarify if the cell deletion occurred equally well in all populations of adipocytes.

Thank you for your interest. We have now tested the expression of 20 genes in total to examine the expression of adrenergic receptors, lipases, adipokines, adipogenic transcription factors, diphthamide biosynthesis enzymes, and other factors (these updated results are presented in Figures 7 and 9 of the main manuscript). Since BMAT volume is extremely small in mice, it takes about 8 to 10 mice to get one BMAT RNA sample for qPCR analysis, limiting the number of analyses that we can perform. Due to limitations and challenges associated with gene studies – we also performed in vivo functional analyses (fat transplant, temperature challenge, and β3-agonist stimulation). Altogether, this work identified resistance to adrenergic-mediated lipolysis, increased lipid storage, and decreased adipokine expression as the main differences between FF and control bone marrow adipocytes. Using the newly added lineage tracing model, we also show that most of these ectopic FF adipocytes are derived from adiponectin-negative progenitor cells. In our working model, we propose that these may be Osteo-CAR cells. Since peripheral adipose tissue does not have such a niche, this may explain why adipocytes are not preserved in any peripheral depots in FF mice. In response to the reviewers, we have also added an additional statement in the third paragraph of the discussion to comment on the specificity of this secondary adipogenesis pathway to BMAT:

“Because peripheral adipose tissues do not have such an alternate progenitor population within their niche, this may explain why this secondary adipogenesis pathway in FF mice is unique to the bone marrow and is absent in other depots including WAT and BAT.”

Regarding clarification of cell deletion in extramedullary populations of adipocytes, we present gross pictures (Figure 2), histological images (Figures 2, 5, and Figure 5—figure supplement 1), and tissue masses (Figure 8—figure supplement 3) confirming the absence of both white and brown adipocytes at diverse extramedullary depots in FF mice (inguinal, gonadal, perirenal, intrascapular, periarticular in the knee joint, and ‘mechanical’ subcutaneous fat normally found in the feet and around the tail). In addition, we were unable to detect circulating adiponectin, supporting the efficacy of the DTA-induced cell death.

– The secondary adipogenesis pathway may in fact be relevant only in extreme circumstances such as in congenital generalized lipodystrophy. Also, the "bone-specific adipogenesis pathway" has been documented only for adiponectin-free fat-free mice. The authors should therefore add "in fat-free mice" at the end of the title, and revise the abstract which appears to be oversimplified and overstated and the conclusions some of which are not supported by experimental evidence.

Thank you for this point. We have modified the title and abstract as indicated in the response to comment #1. We have also added a limitations section to highlight the use of the fat-free model.

– Similarly, the stated hypothesis is not really a hypothesis, because this point has been demonstrated in a number of previous studies. The real novelty is that adiponectin-negative bone marrow adipocytes appear under special and rather extreme circumstances. It would be essential to demonstrate that these bone marrow adipocytes in FF mice are indeed adiponectin-negative (by immunostaining).

We have tried two adiponectin antibodies with adiponectin-knockout tissue as a negative control. Unfortunately, despite careful optimization, we were not able to obtain a specific signal in either bone/BMAT or adipose tissues. Given this, we are not able to include immunostaining in this manuscript. However, to address this point, we have now included data from the triple mutant FF^mTmG^ lineage tracing reporter mice (see response to point #2 above). In addition to this new lineage tracing data, we also found that gene expression of *Adipoq* was reduced in purified BMAds (Figure 9) and that, despite maintenance of BMAds, FF mice did not have circulating adiponectin (Figure 2). Altogether, these points support the conclusion that these ectopic adipocytes are largely adiponectin^-/low^.

– The authors do not consider the possibility that DTA does not kill all adiponectin-expressing cells. The DTA allele is known to be ineffective and insensitive to Cre recombinations. This is important as leakiness could undermine the major conclusions of this study. The authors should therefore determine the efficiency of cell ablation, by combining DTA and mTmG alleles with Adipoq-Cre. Any adiponectin-expressing cell escaping DTA should turn GFP+.

Thank you for this suggestion, this is an important point. We have now included data from the triple mutant fat-free FF^mTmG^ lineage tracing reporter mice to determine the efficiency of the DTA-mediated cell ablation as well as to explore the origins and molecular features of this unique BMA population (see response to point #2 above).